# Individual Tree Crown Segmentation Directly from UAV-Borne LiDAR Data Using the PointNet of Deep Learning

Xinxin Chen [1], Kang Jiang [1], Yushi Zhu [1], Xiangjun Wang [2] and Ting Yun [1,3,*]

1   College of Information Science and Technology, Nanjing Forestry University, Nanjing 210037, China; cxx19950702@gmail.com (X.C.); jiangkang@njfu.edu.cn (K.J.); iszhuyushi@gmail.com (Y.Z.)
2   Rubber Research Institute, Chinese Academy of Tropical Agricultural Sciences, Haikou 571101, China; hnwxj@catas.cn
3   Co-Innovation Centre for Sustainable Forestry in Southern China, Nanjing Forestry University, Nanjing 210037, China
*   Correspondence: yunting@njfu.edu.cn; Tel.: +86-025-8542-7464

**Abstract:** Accurate individual tree crown (ITC) segmentation from scanned point clouds is a fundamental task in forest biomass monitoring and forest ecology management. Light detection and ranging (LiDAR) as a mainstream tool for forest survey is advancing the pattern of forest data acquisition. In this study, we performed a novel deep learning framework directly processing the forest point clouds belonging to the four forest types (i.e., the nursery base, the monastery garden, the mixed forest, and the defoliated forest) to realize the ITC segmentation. The specific steps of our approach were as follows: first, a voxelization strategy was conducted to subdivide the collected point clouds with various tree species from various forest types into many voxels. These voxels containing point clouds were taken as training samples for the PointNet deep learning framework to identify the tree crowns at the voxel scale. Second, based on the initial segmentation results, we used the height-related gradient information to accurately depict the boundaries of each tree crown. Meanwhile, the retrieved tree crown breadths of individual trees were compared with field measurements to verify the effectiveness of our approach. Among the four forest types, our results revealed the best performance for the nursery base (tree crown detection rate $r = 0.90$; crown breadth estimation $R^2 > 0.94$ and root mean squared error (RMSE) < 0.2m). A sound performance was also achieved for the monastery garden and mixed forest, which had complex forest structures, complicated intersections of branches and different building types, with $r = 0.85$, $R^2 > 0.88$ and RMSE < 0.6 m for the monastery garden and $r = 0.80$, $R^2 > 0.85$ and RMSE < 0.8 m for the mixed forest. For the fourth forest plot type with the distribution of crown defoliation across the woodland, we achieved the performance with $r = 0.82$, $R^2 > 0.79$ and RMSE < 0.7 m. Our method presents a robust framework inspired by the deep learning technology and computer graphics theory that solves the ITC segmentation problem and retrieves forest parameters under various forest conditions.

**Keywords:** deep learning; individual tree crown segmentation; Airborne LiDAR data; computer graphics



## 1. Introduction

The accurate separation of individual trees plays an essential role in the tree parameter inversion. Forest parameters [1], such as tree location, tree height, canopy density, crown width, tree species, and diameter at breast height (DBH), are crucial for forest resource management, field inventory retrieval, and silvicultural activity execution [2]. The traditional acquisition of tree structural parameters was usually through field measurements, but this process is extremely time-consuming, labor-intensive, and destructive [3]. Light detection and ranging (LiDAR) is an active remote sensing technology, as its high precision and high efficiency has led to it becoming one of the most efficient surveying techniques for acquiring detailed and accurate target phenotypic data [4]. In terms of the carrying platform, laser scanning systems can be classified into four categories: airborne laser scanning

(ALS) [5], satellite-based laser scanning (SLS) [6], vehicle-borne laser scanning (VLS) [7], and terrestrial laser scanning (TLS) [8]. Similar like ALS, the unmanned aerial vehicle (UAV) provides an alternative platform for lidar data acquisition, which can decrease the cost and provide denser LiDAR points when flying at a slow speed and a lower altitude [9].

As mentioned above, the detection and the segmentation of a single tree crown is a fundamental step to accurately estimate the individual tree structural attributes [10]. We classified the existing methods of individual tree crown (ITC) segmentation into two main categories, which are widely used in the field of forestry: (1) the canopy height model (CHM)-based approach [11] which uses image processing to segment a single canopy and then uses a local maximum to define the location of the treetop. Algorithms such as the marker-controlled watershed algorithm [12], graph-based segmentation algorithm [13], and localized contour expansion based on the topological relationship [14], have also been adopted to accomplish tree crown segmentation based on the detected treetop locations. Nevertheless, the relatively low accuracies for these algorithms are always caused by the inhomogenous, interlocked, and blocked canopies [15]. (2) The point-based approach is a method that requires massive computation of 3D points. This method can effectively reduce the loss of information at the tree level [16] and avoid errors caused by the point cloud interpolation during the process of generating CHM, such as the K-means clustering [17], mean-shift algorithm [18], voxel space projection [19], adaptive multiscale filter [3], and regional growth method [20]. However, for natural forests in which tree crowns can be extremely irregular and are often heavily intersected, the results of accurate individual tree crown segmentation by these methods still need to be improved.

Deep learning, as a new area of machine learning, has been widely used in image classification, object detection and localization among other aspects [21]. Deep learning algorithms using Convolutional Neural Networks (CNN) have shown encouraging results for the automatic classification of two dimensional (2D) images [22], such as facial recognition [23], autonomous driving [24], medical imaging [25], and fruit and vegetable detection [26,27]. However, the phenotypic structure [28] of more 3D objects is directly reflected in the point cloud, and the original information and spatial characteristics will be lost if a 2D network is used. Therefore, 3D object detection was proposed by many research communities.

At present, with the development of Laser Scanning technology, 3D deep learning has received great attention. The methods of 3D point cloud recognition based on deep learning can be divided into four categories: (1) A feature-based method [29] which extracts feature descriptors from the point clouds and then uses a fully connected network to classify the shape. However, this method is constrained by the representation power of the features extracted; (2) The multi-view method [30] which applies 2D convolutional network to classify the 2D images that use a projection strategy to convert 3D point clouds or shapes from different perspectives. The method based on multi-view achieves good performance in classification tasks [31], but it loses the original 3D spatial position information in the process of being transformed into 2D image; (3) The method based on voxelization which converts the unordered point clouds to a continuous arrangement of the voxel grid and classifies the voxel grid by 3D convolutional neural network [30,32]. The method based on voxelization can be effectively retained the original spatial information of the point clouds in each voxel, which is beneficial for subsequent refinement processing for the accurate target depiction. Compared the performances of the aforementioned three methods, the voxel-based method using the divide-and-conquer strategy [33] to recognize the small targets from the whole collected data of the studied complex scenes and then stitched the recognized results together to realize the small objects extraction from the whole collected data. Many researchers have proposed some related deep learning frameworks, e.g., PointNet [34], Kd-Network [35], and PointCNN [36]. PointNet was the pioneering work with raw point clouds in each voxel as input for deep learning. The model PointNet provides a unified architecture for applications ranging from object classification, part segmentation, to scene semantic parsing.

In this paper, a novel individual tree segmentation method combined with a Point-Net method is proposed. The research objectives of this paper mainly include (1) using UAV-borne laser LiDAR to collect data; (2) voxelizing the training and testing sites; (3) transforming the data of the training and testing sites from voxelization into the format required by the PointNet for training and testing; and (4) identifying the segmented voxels based on the PointNet and using the gradient information to construct and describe the boundary of the tree in each voxel to realize the individual tree crown (ITC) segmentation. The workflow of our method is shown in Figure 1.

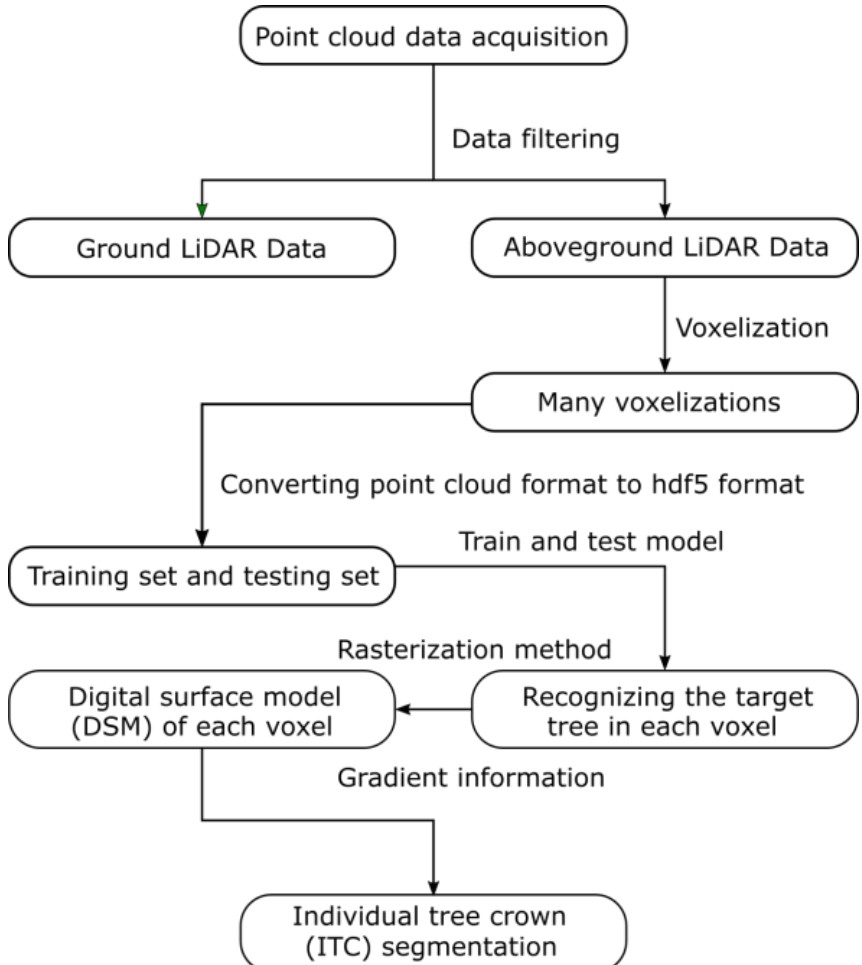

**Figure 1.** The main steps for our individual tree segmentation based on the deep learning method.

## 2. Materials and Methods

### 2.1. Study Area

The study area is located in Qishan scenic area of Chizhou City (30°38′15.89″ N, 117°30′11.33″ E), southwest of Anhui province (Figure 2), China. As a national forest city, Chizhou city has a warm climate with four distinct seasons and abundant rainfall, belonging to a warm and humid subtropical monsoon climate. The average annual precipitation here is 1400 to 2200 mm, the average annual temperature is 16.7 °C, and the average temperatures in the coldest month (January) and the hottest month (July) are approximately 3.1 °C and 28.7 °C, respectively. Qishan covers a total area of 36 km², with the highest elevation of 868 m. The tree population in this area mainly consists of 10 different tree species, including metasequoias (*Metasequoia glyptostroboides Hu & W. C. Cheng*), Chinese firs (*Cunninghamia lanceolata (Lamb.) Hook*), cedars (*Cedrus deodara (Roxb.) G. Don*), ginkgoes (*Ginkgo biloba L.*), sapindus (*Sapindus mukorossi Gaertn.*), apple trees (*Malus pumila Mill.*), poplars (*Populus L.*), camphors (*Cinnamomum camphora (L.) Presl*), ceibas (*Bombax malabaricum*) and locust trees

(*Sophora japonica Linn.*). As shown in Figure 2, four experimental site types, including nursery base (experimental site 1), the monastery garden (experimental site 2), the mixed forest (experimental site 3) and the defoliated forest landscape (experimental site 4), in the Qishan scenic were chosen for our experiments. Moreover, experimental sites 1, 2, and 4 are located at the foot of the mountain, while experimental site 3 is located at the waist of a mountain with unevenly hilly terrain. Four experimental sites consisted of buildings, shrubs, and trees.

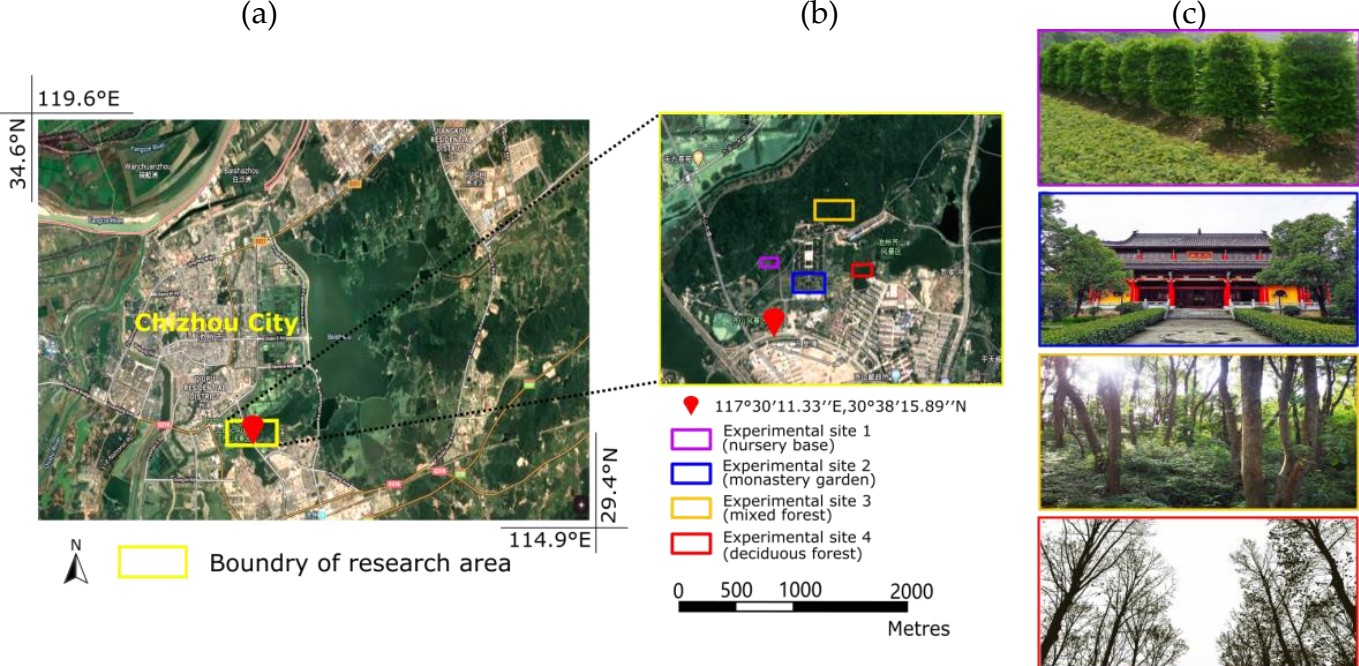

**Figure 2.** General situations of the study area. (**a**): The location of the study area and the four experimental sites within the Qishan scenic area, Chizhou, Anhui province, China. (**b**): The remote sensing image acquired from Google Earth, where the different coloured rectangles mark the edges of the different experimental sites. (**c**): The photos show the growth of trees in the four experimental sites.

Four subsets extracted from the four experimental sites occupying the area of 1947.16, 44,596.64, 60,601.78 and 14,780.11 square meters, respectively, were taken as the study area in the follow-up experiments. The vegetation components and buildings in 50% of the area of each subset were extracted as the training samples. The rest of the four subsets served as testing samples (which did not intersect with the subset used as the training samples).

*2.2. Laser Data Acquisition*

The LiDAR data were measured using a Velodyne HDL-32E sensor on the DJI FC6310 unmanned aerial vehicle (UAV) [37]. The laser group in the system can realize the adjustment of angle from −30.67° to +10.67° and provide a 360° horizontal field of view. The sensor can output approximately 700,000 scan point clouds per second with a measurement accuracy of ±2 cm. In addition, this sensor has the advantages of penetrating smoke and fog, and the working environment can be from −10 °C to +60 °C, which greatly improves the redundancy of working environment. The Velodyne LiDAR system integrates laser scanning with SLAM (simultaneous localization and mapping) technologies [38] to rapidly complete the registration of each scan and generate a high-density point cloud for each target tree. During data acquisition, the flight speed, flight altitude, and laser scanning overlap were set as 18 m/s, 60 m (above the take-off location level) and 40%, respectively. The final extracted point clouds were stored in LAS 1.2 format. The average point density of the collected LiDAR data for the nursery base, monastery garden, mixed forest and

defoliated forest habitats were 1511.30 pts m$^{-2}$, 1002.17 pts m$^{-2}$, 722.31 pts m$^{-2}$, and 502.34 pts m$^{-2}$, respectively.

### 2.3. Data Pre-Processing

After acquiring the point cloud data from the experimental sites scanned by the laser scanner, we used the method of Gaussian filtering [21] to remove noise points from the scanned data. The point clouds after denoising were classified as aboveground points and ground points using the cloth simulation filtering (CSF) [39] method. Then, the aboveground points were voxelized according to different voxel sizes and points within a voxel were randomly sampled to 1024 points. We converted the point clouds in each voxel constituting the training and testing sets into HDF5 [40] format according to the requirements of the PointNet. In this experiment, the criteria of the HDF5 file included two parts: data and labels. In the data section, data converted from the scanned points as training and testing sites were an array of $n \times 1024 \times 3$, where $n$ represents the total number of segmented input voxels; 1024 represents the number of point clouds of random sampling in a voxel and 3 represents the dimension, i.e., spatial position (x, y, z). Labels were used to identify certain properties or features, or classifications or contained objects.

### 2.3.1. Training Data

In this study, we manually generated three types of the training data, which include: (1) individual trees belonging to a variety of tree species and under two plant physiological status (with and without leaves), (2) different Chinese architectural styles, such as, palaces, city walls, temples, and houses, and (3) other objects including bare ground, understorey vegetation and a small portion of point clouds regarding a single tree (usually <20%) or intersecting parts of adjacent trees. The number of training samples (trees and buildings) for the nursery base, monastery garden, mixed forest plot and defoliated forest landscapes were 501 (trees), 168 (trees)/334 (buildings), 426 (trees), and 166 (trees), respectively. Figure 3 shows the partial training data, where manually extracted point clouds of the individual trees or part of the buildings were bounded in a voxel.

A large number of samples is the basis for high-precision training, so it is worth having as much training data as possible to train neural networks to avoid over-fitting. In our study, data augmentation [41] was used to solve this problem. The method of data augmentation is a strategy that increases the diversity of data available for training models without actually collecting new data, thus improving the accuracy of the model. We generated new training data set based on the rotation of the entire point cloud in each voxel by a random angle and along the vertical axis. Meanwhile, the strategy of moving every point in each voxel with a small offset along a random vector, i.e., jittering the position of each point of every training sample by a Gaussian noise with zero mean and a small standard deviation (ranging 0.02–0.06). As a result, the number of training samples was broadly expanded to 10240.

### 2.3.2. Testing Data

The Four aforementioned experimental sites, i.e., the nursery base, the monastery garden, the mixed forest and defoliated forest were used to test the accuracy and robustness of the method. The number of trees in the testing sites in experimental sites 1, 2, 3 and 4 was 522, 160, 456, and 167, respectively (Table 1). After removing the noise points, the four scanned point sets $V_1$, $V_2$, $V_3$, $V_4$ of the corresponding experimental sites were subdivided into many voxels through voxelization. Then, the point cloud in each voxel was obtained by each voxelization (i.e., $v_j$, $v_j \in V$) according to the HDF5 criteria.

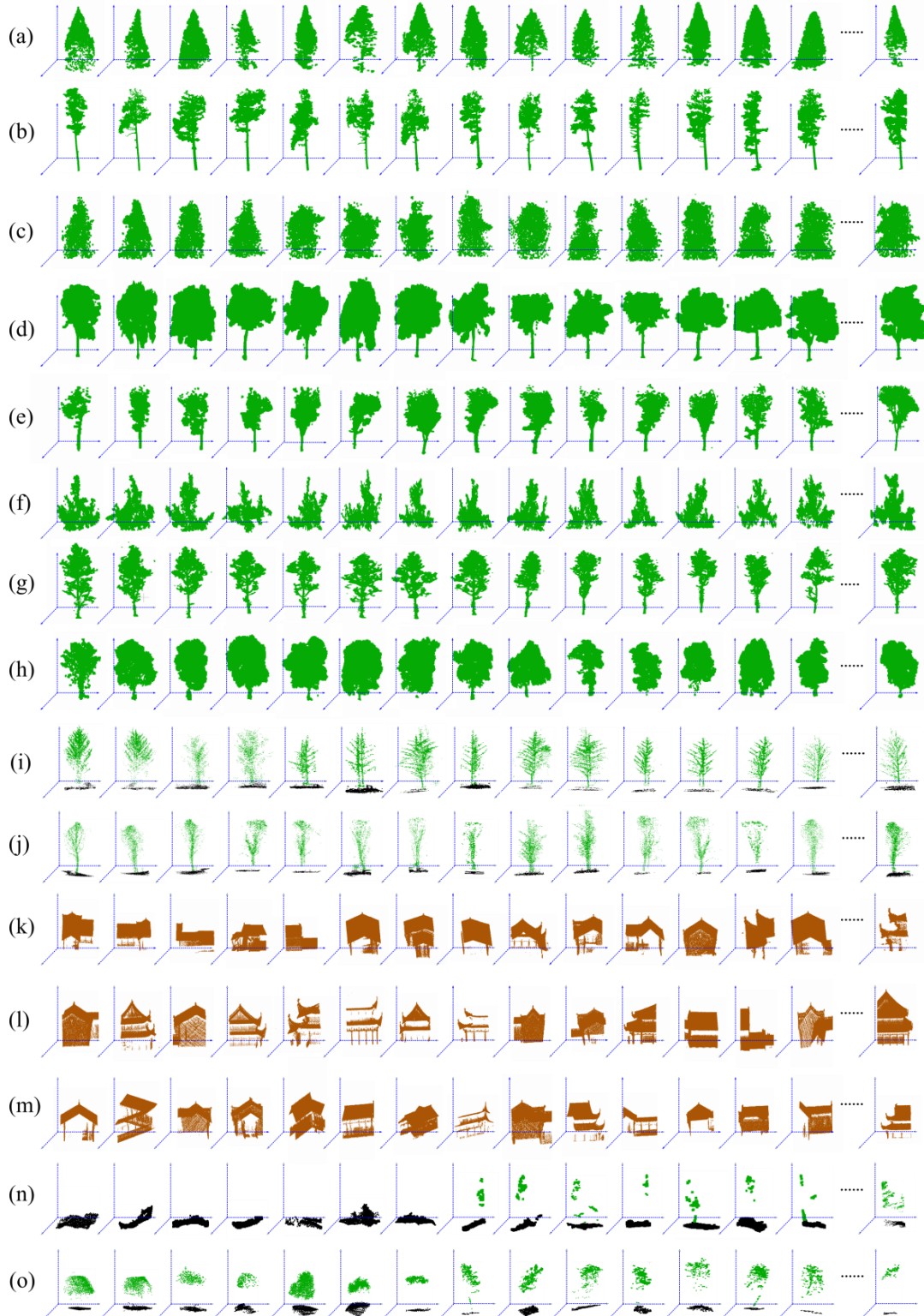

**Figure 3.** Illustration of the partial training sets of point clouds collected for the PointNet network. Lines 1 to 10 are extracted individual trees of corresponding scanned points regarding different tree species, namely (**a**) metasequoias, (**b**) Chinese firs, (**c**) cedars, (**d**) ginkgoes, (**e**) sapindus, (**f**) apple trees, (**g**) poplars, (**h**) camphors, (**i**) ceibas, and (**j**) locust trees, respectively. The last five lines (**k**–**o**) are parts of buildings including palaces, city walls, temples, and houses in different Chinese architectural styles and other objects including bare ground, understorey vegetation and a small portion of point clouds regarding a single tree (usually <20%) or intersection parts of adjacent trees.

**Table 1.** A detailed description of the dataset for our deep learning method.

|  |  | Nursery Base | Monastery Garden | Mixed Forest | Defoliated Forest |
|---|---|---|---|---|---|
|  | NT | 1059 | 336 | 921 | 338 |
|  | NP | 2,942,740 | 44,693,237 | 43,773,108 | 7,424,662 |
|  | NPPT | 2779 | 31,020 | 47,528 | 21,966 |
|  | Area (m$^2$) | 1947.16 | 44,596.64 | 60,601.78 | 14,780.11 |
| Training sites | NT | 537 | 176 | 465 | 171 |
|  | NP | 1,485,416 | 22,432,004 | 22,044,103 | 3,722,314 |
|  | Area (m$^2$) | 984.13 | 22,377.18 | 30,461.12 | 7439.57 |
| Testing sites | NT | 522 | 160 | 456 | 167 |
|  | NP | 1,457,324 | 22,261,233 | 21,729,005 | 3,702,348 |
|  | Area (m$^2$) | 963.03 | 22,219.46 | 30,140.66 | 7340.54 |

NT: the number of trees. NP: the number of scanned points. NPPT: the average number of scanned points per tree.

### 2.4. Training by PointNet

PointNet is the first deep neural network that directly processes out-of-order point cloud data. The PointNet has three core building blocks, i.e., the transformation networks (T-Net), the max pooling layer as a symmetric function to aggregate information from all the voxels and the multi-layer perceptron (MLP) network. A point cloud $p_i^j(x_i, y_i, z_i)$ is represented as a 3D scanned point in the *j*-th voxel belonging to the scanned point set $P \subset R^3$, where each point $p$ is a vector of its $(x, y, z)$ coordinate as point's channels. There are three core properties for the point cloud, including (1) being unordered, which represents a network that consumes N 3D point sets that needs to be invariant to N! permutations of the input set in data feeding order, (2) the interaction among points, which means that points are not isolated, and neighbouring points form a meaningful subset, and (3) invariance under transformations [42], which represents that the learned representation of the point set should be invariant to certain transformations. Therefore, it is necessary to design a symmetric function in algebraic combinatorics, of which the value is independent of the order in the scanned points in a voxel. The PointNet network is represented by the symmetric Equation (1).

$$f(p_1^j, p_2^j, \ldots, p_i^j, \ldots, p_{1024}^j) = \gamma(\max_{i=1,\ldots,1024}\{h(p_i^j)\}) \tag{1}$$

In the formula, $p_1^j, p_2^j, \ldots, p_i^j, \ldots, p_{1024}^j$ is the input disordered point cloud in the *j*-th voxel; $p_i^j \in P$; 1024 is the number of input point clouds for each voxel; *f* is the continuous set function and map a set of points to a vector; $\gamma$ represent the multi-layer perceptron network and *h* represents the composition of a single variable function and a max pooling function. The values of the continuous set function *f* in Equation (1) are invariant regardless of the input order of the point cloud.

Figure 4 shows the network architecture of PointNet. The input of the network was the three-dimensional coordinates ($n \times 1024 \times 3$) of the three-dimensional point cloud containing *n* voxels and 1024 points within a voxel. T-Net is a mini-network that can predict the affine transformation matrix. The first T-Net in the network generated an affine transformation matrix to normalize the rotation, translation and other changes of the point cloud. At this time, the input of the first T-Net was the original point cloud data, and the output (aligned data) of the first T-Net was a $3 \times 3$ rotation matrix. Then, the original 3D point data was multiplied by the transformation matrix ($3 \times 3$) learned by the first T-Net to achieve data alignment for ensuring the invariability of the model for specific spatial transformation. The aligned data of point clouds ($1024 \times 3$) in each voxel was passed through a multi-layer perceptron (MLP (64, 64)) with the given numbers of layer sizes shown in the bracket to obtain the matrix ($1024 \times 64$). The fully connected layers of MLP are shown by the three dotted boxes in the upper part of Figure 4. After that, 64-dimensional features

were extracted for each voxel, and then the $64 \times 64$ transformation matrix was predicted by the second feature space transformation matrix of the T-Net prediction network, which was applied to the features to achieve feature alignment. Similarly, the matrix ($1024 \times 64$) was multiplied by the transformation matrix ($64 \times 64$) to achieve the alignments of features. Then, the second MLP (64,128,1024) was used for feature extraction based on each voxel until the feature's dimension is changed to 1024, and then the global feature vector of each voxel was extracted by max pooling layer. Finally, the global features of the $1 \times 1024$ dimension pass through the third MLP (512, 256, 3), resulting in 3 classifications, where 3 represents the categories of the classification (i.e., the number of categories defined by the label, 0 represents the tree, 1 represents the building and 2 represents other objects). Each category corresponds to the classification scores for the point cloud. Then, through the activation layer based on the SoftMax function, the prediction probability of the point clouds in each voxel can be obtained.

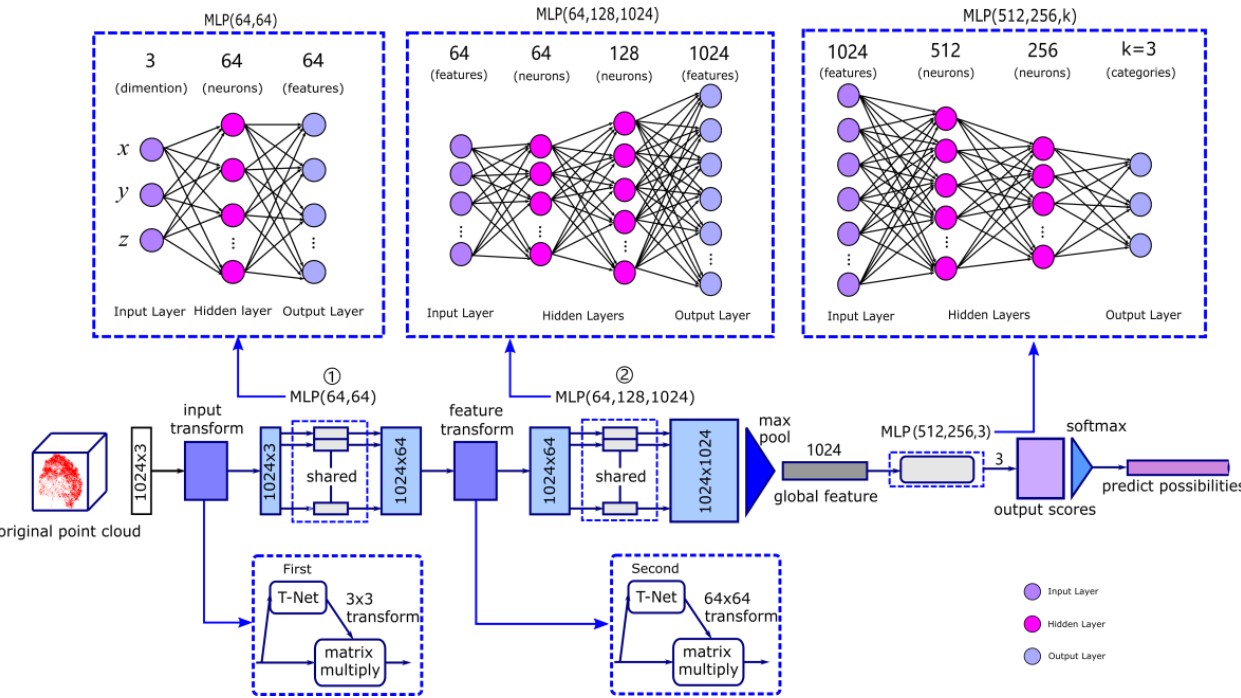

**Figure 4.** The architecture of PointNet. The architecture is mainly composed of two transformation matrix prediction networks (T-Net), three multi-layer perceptrons (MLP) and a max pooling layer. The network takes 1024 points in a voxel as input, applies input and feature transformations, and then aggregates point features by max pooling. The output is the predicted possibilities for the classifications of classes.

### 2.5. The loss Function of the Training Process

The SoftMax cross-entropy function (Formula (2)) was taken as the loss function for the deep learning network. In the training process, the loss function is defined as follows:

$$Loss = -\frac{1}{k+1}\sum_{l=0}^{k} L_l + weight_{regre} L_{reg} = -\frac{1}{k+1}\sum_{\varsigma}\sum_{l=0}^{k}\left(indic_{\varsigma,l} \bullet \log\left(\hat{y}_l^j\right)\right) + weight_{regre} L_{reg} \tag{2}$$

$$indic_{\varsigma,l} = \begin{cases} [0,0,1] & The\ first\ category\ \zeta = 0 \\ [0,1,0] & The\ \sec ond\ category\ \zeta = 1 \\ [1,0,0] & The\ third\ category\ \zeta = 2 \end{cases} \tag{3}$$

$$\hat{y}_l^j = soft\max\left(Z_l^j\right) = \frac{e^{Z_l^j}}{\sum_{l=0}^{2} e^{Z_l^j}} \tag{4}$$

$$Z_l^j = \omega * p^j \tag{5}$$

$$L_{reg} = l2\_loss\left(I - AA^T\right) \tag{6}$$

In Formula (2), $indic_{\varsigma,l}$ represents an indicator related to the number of classifications. If the calculated category $\zeta$ and the current category $l$ of the voxel $j$ are the same, the indicator is assigned to 1, otherwise equal to 0. In our work, the total number of the categories is 3. Hence, $k= 2$ and $l = \{0, 1, 2\}$. $\bullet$ represents the dot product of two matrices. $L_{reg}$ is used to constrain the feature transformation matrix, where $A$ is the feature alignment matrix (i.e., the transform matrix $64 \times 64$ obtained from the second T-Net), and $I$ is a unit matrix. The function $l2\_loss$ represents the sum of the square of each element in the matrix and then divided by 2. Here, the value of $weight_{regre}$ is set to 0.001. $\hat{y}_l^j \in [0, 1]$ is the probability of the network output for the $j$-th voxel which uses the SoftMax function, indicating the probability that the input voxel belongs to the $l$-th category. $Z_l^j$ is the calculated probability value of the point clouds in the $j$-th voxel belonging to the $l$-th category after neural network analyzing. $\omega$ is the linear weights of the network model and $p^j$ is the point cloud of the $j$-th voxel of the segmented input voxel.

The weight ($\omega$) of each layer of the deep convolution neural network is updated by a stochastic gradient descent (SGD) algorithm [43]. A layer is a container that usually receives weighted input, transforms it with a set of mostly non-linear functions and then passes these values as output to the next layer. When the training loss function is less than a certain loss threshold value (i.e., convergence), then the training is stopped and the weight of each layer of the fixed network is no longer changed, so that the trained deep convolutional neural network can be obtained.

*2.6. Individual Tree Segmentation*

The testing process included the following steps. The point clouds of each testing site were assigned to continuous distributed voxels by voxelization. Then, the subdivided point clouds in each voxel were analyzed by the PointNet framework with the learned parameters through the training stage and the classification results of each voxel were obtained. For the point clouds in a voxel recognized as trees, we refined tree crown boundary delineation based on the height-related gradient information and accurate depicted the crown boundaries beyond the limitation of the defined voxel boundaries.

First, the point clouds in a voxel classified as the categories of trees were mapped into evenly distributed planar raster $C$ of digital surface model (DSM) [44]. The elevation value of a raster cell $c_k \in C$, $k = 1, 2, \ldots m^2$ was equal to the largest height value of the points within the cell, where $m^2$ represents number of the cells contained in a raster derived from the point clouds within a voxel.

Then, a local maximum searching algorithm [45] was adopted to find the positions of treetop in each voxel. The Hamiltonian operator, denoted hereafter by $\nabla$, represents the gradient of the cell in the three-dimensional space which was defined by $x$, $y$ (horizontal) and $z$ (vertical) axes. The corresponding Equation is as follows:

$$\nabla c_{r,p,q} = \frac{\partial C}{\partial x}\vec{u} + \frac{\partial C}{\partial y}\vec{v} + \frac{\partial C}{\partial z}\vec{w} \tag{7}$$

In Equation (7), the $\vec{u}$, $\vec{v}$ and $\vec{w}$ are the unit vectors in the $x$, $y$ and $z$ directions, respectively. The gradient is the result of the Hamiltonian operator directly acting on the DSM $C$ of each voxel. In our study, the DSM of each voxel at a raster cell resolution is $11 \times 11$. $\frac{\partial C}{\partial x}$, $\frac{\partial C}{\partial y}$ and $\frac{\partial C}{\partial z}$ are the derivatives of the highest scanned point in each raster cell along the $x$, $y$ and $z$ directions, respectively. The phenotypic features of the tree crown periphery present a downward hierarchical structure, i.e., the height value on the surface pixels of the crown decreases gradually from the peak to the surroundings. Hence, there must be saddle points (the lowest point and the gradient of this point close to 0) existing between two adjacent trees. Assisted with the calculated gradient information of each

cell, the valley line between adjacent tree crowns was located by contour line extraction method [46], which is similar to the graph cut method based on the node depth. Finally, if the height value of part of the point clouds within the two adjacent voxels with a continuous downward trend, i.e., continuous gradient descent along the similar directions, it indicates the point clouds belonging to the same tree crown were subdivided into two parts by voxelization. Hence, the two parts of one tree crown should be merged (Figure 5).

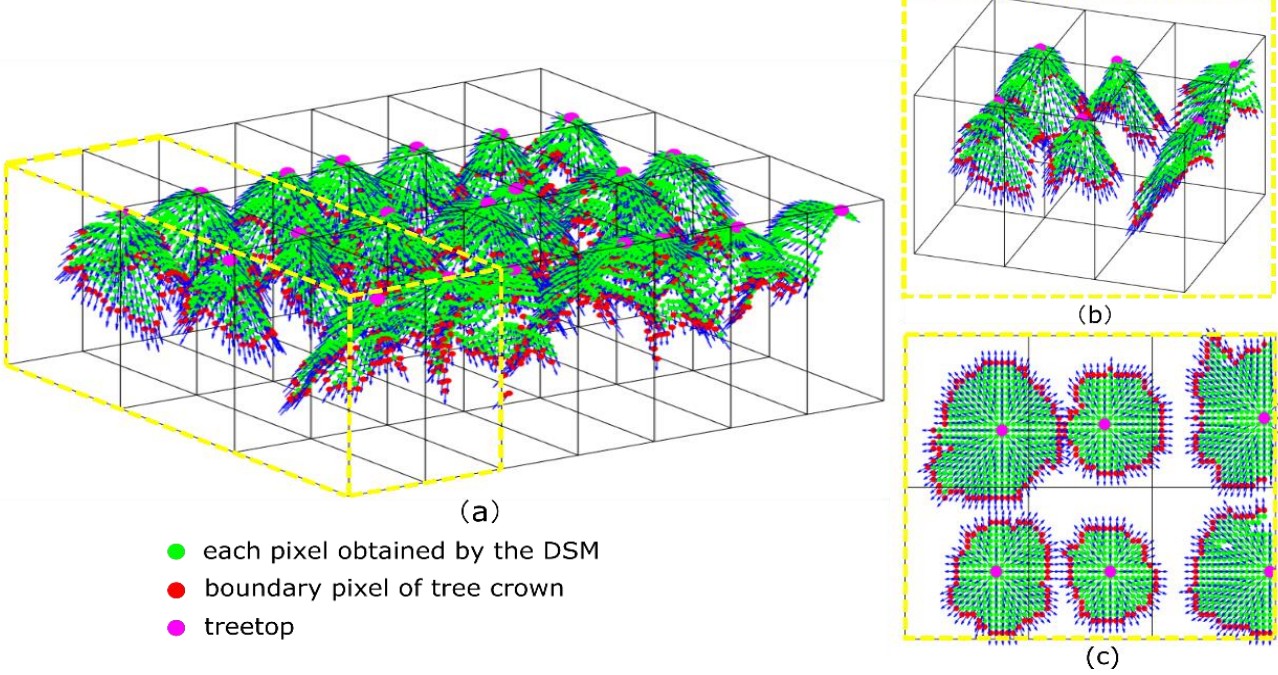

● each pixel obtained by the DSM
● boundary pixel of tree crown
● treetop

**Figure 5.** Schematic diagram showing the individual tree crown segmentation for the point clouds in the 21 adjacent voxels based on height-related gradient information. The black cuboid represents the segmented voxel classified as a tree by the PointNet. (**b**,**c**) are the side and top views of the zoomed area of the yellow cuboid shown in figure (**a**), respectively.

The segmentation results of the selected testing sites with different forest stand structural features were evaluated versus the manually measurement results. *TP* (true positive) represents the number of trees that were correctly segmented. *FN* (false negative) represents the number of segmented trees that were not detected (omission error). *FP* represents (false positive) the number of segmented trees that do not exist in reality but were incorrectly added (commission error) by our model. In addition, the *r* (recall), *P* (precision), and *F* (*F*-score) for the three testing sites were calculated using the following Equations [47]:

$$r = \frac{TP}{TP + FN} \tag{8}$$

$$P = \frac{TP}{TP + FP} \tag{9}$$

$$F = 2 * \frac{r * P}{r + P} \tag{10}$$

where the *r* represents the detection rate of the tree, *P* represents the correctness of the detected trees, and *F* represents the overall accuracy of the detected trees. As can be seen in the formulas, high *TP*, low *FN*, and low *FP* values represent high accuracy of the tree detection.

## 3. Results

### 3.1. Results of Training and Testing of the PointNet Model

The experiments except for a section about deep learning performed on a windows 10 64-bits PC equipped with an Intel(R) Core (TM) i7-7700 CPU @2.80 GHz processor (Intel Inc., Santa Clara, CA, USA), and 16GB-RAM. Since deep learning involves automating a computer system to study a large amount of training data and requires high computing power, we used NVIDIA RTX 2080Ti GPU (NVIDIA Inc., Santa Clara, CA, USA) instead of CPU to reduce our training time. In the model of PointNet, the learning rate is 0.0001, the batch size is 16, and the number of epochs is 200. The training loss and training accuracy are plotted in Figure 6. The total training and testing time is approximately 100 h.

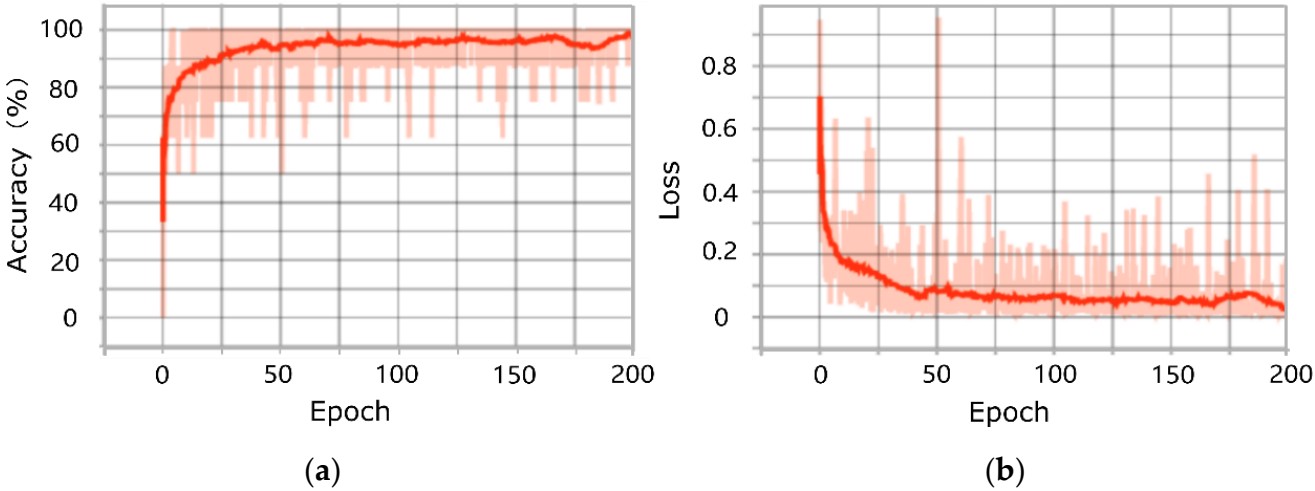

**(a)**          **(b)**

**Figure 6.** (**a**,**b**) are the curves of training accuracy value and training loss value of PointNet for tree recognition from input voxels. Fluctuations in the light colored region were caused by repeatedly learning effective features from complicated samples in a batch to identify whether the voxel is a tree, but the overall upward trend and the downward trend of the curve indicates a better convergence result of training.

With the continuous epoch of the learning process, training samples (point cloud in each voxel) showed an increasing trend for training accuracy and a decreasing trend for training loss, indicating that our PointNet was a global optimization process. Both training accuracy and training loss show a significant increase and decrease in the first 25 epochs, respectively. The reason is likely that when dealing with samples overwhelming in 3D object classification, the model PointNet shows incompatibility due to the fact that its gradients are mainly determined by these easy-classified samples. During the training process, the neuron network encountered some complicated samples in a batch, e.g., a voxel containing parts of multiple trees, small proportion of an individual tree data or some dwarf shrubs, which impaired the learning efficacy of the model and resulted in the strong fluctuations in the value of the regression loss function. After 75 epochs, accuracy and loss of training sample converged to 0.96 and 0.009, respectively, indicating the strong fitting ability of the PointNet. Figure 7 shows the side view of the recognition results of the four testing plots obtained by the model of the PointNet.

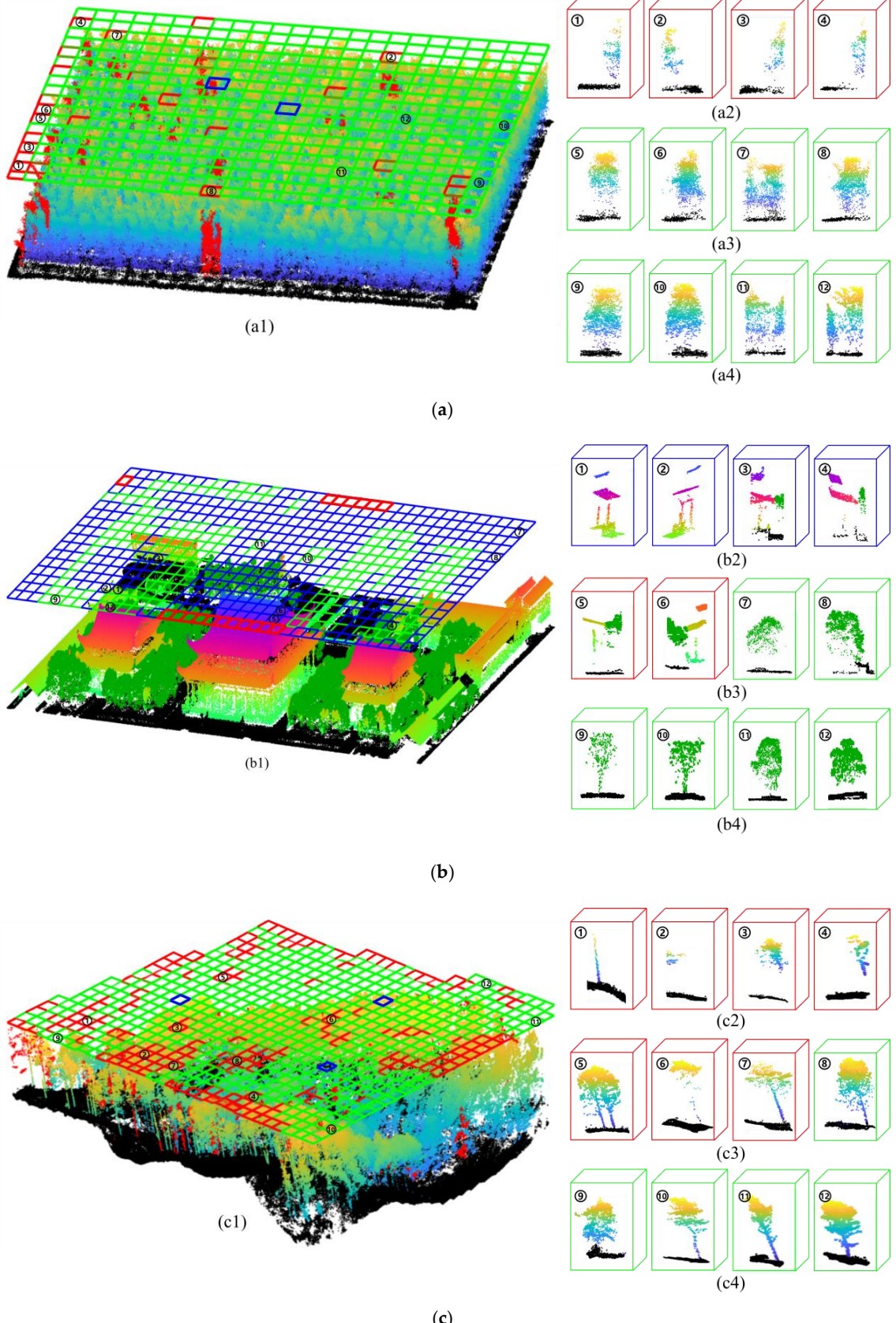

**Figure 7.** *Cont.*

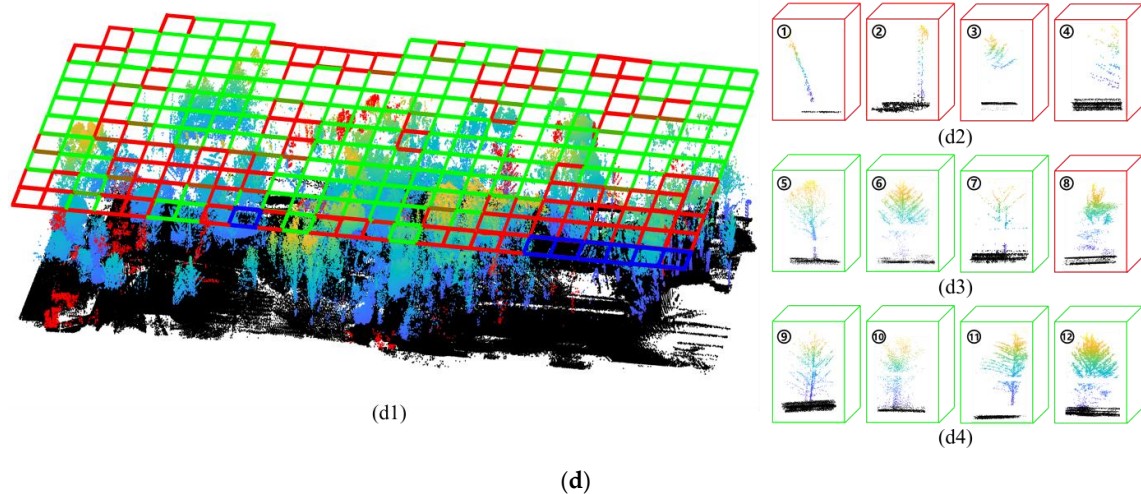

(d)

**Figure 7.** Recognition results of the partial LiDAR data belonging to the four forest plot types in our study sites using Point-Net model: (**a**) nursery base, (**b**) monastery garden, (**c**) mixed forest, and (**d**) defoliated forest. A simplified representation using the upper rectangles in different colours represents the classification results of the point clouds below within each corresponding voxel, where the green, blue and red rectangles in (**a1,b1,c1,d1**) indicate that the point clouds under the rectangle within the voxel were recognized as trees, buildings and other objects, respectively. (**a2–a4,b2–b4,c2–c4,d2–d4**) show the zoomed classification results for the point clouds in some voxels.

The quantitative assessment of individual voxels of four testing sites is listed in Table 2. In the experiment, the setting of voxel size is crucial, which will affect the accuracy of the PointNet model. Therefore, we set different appropriate specifications of the voxels (i.e., the average tree crown width in the E-W and the N-S directions of each testing site were taken as the length and width of the voxel) according to the characteristics (i.e., tree crowns) of trees of the four testing sites as much as possible. The nursery base (testing site 1) has homogeneous forest with similar tree crown sizes, species, and ages. Hence, it is relatively easy to set the size of the voxelization. For the monastery garden (testing site 2) with different types of buildings, different tree species, it will be cumbersome to set the size of the voxel due to the intricate growth and different size of various tree species. For the mixed forest (testing site 3) with various sizes of tree crowns, the complicated intersection of branches and containing roughly 15% of the sub-canopy trees, the defoliated forest (testing site 4) with bare branches, and a few of trees with lower parts covered by surrounding shrubs, it is difficult to ensure that a voxel contains a complete tree.

**Table 2.** Overall accuracy assessment of individual voxels of four testing sites for identifying trees.

|  | Nursery Base | Monastery Garden | Mixed Forest | Defoliated Forest |
|---|---|---|---|---|
| Average tree crown length/width/heights (m) | 1.35/1.36/3.29 | 6.46/5.81/6.34 | 7.08/6.59/13.71 | 5.23/5.2/14.95 |
| Length/width/height of voxels (m) | 1.35/1.36/4.92 | 6.46/5.81/26.96 | 7.08/6.59/48.06 | 5.23/5.2/20.96 |
| Total number of voxels after voxelization | 528 | 592 | 646 | 270 |
| Identification results | T:497; B:5; O:26 | T:168; B:331; O:93 | T:424; B:30; O:192 | T:165; B:19; O:86 |

T: the number of voxels identified as trees. B: the number of voxels identified as buildings. O: the number of voxels identified as other objects.

The average tree crown size obtained by preliminary forest survey was used to define the size of the voxels. Here, we defined voxels with lengths, widths and heights of 1.35 m, 1.36 m, 4.92 m for the nursery base, 6.46 m, 5.81 m, 26.96 m for the monastery garden, 7.08 m, 6.59 m, 48.06 m for the mixed forest, and 5.23 m, 5.2 m, 20.96 m for the defoliated forest, respectively.

For four testing sites, nursery base, monastery garden, mixed forest and defoliated forest, the identified voxels of trees are 470, 136, 365, and 137, respectively. For the nursery base, as shown in Figure 7(a2), the main errors appeared when the voxel containing scanned points was regarded as a small portion of tree saplings with immature tree crown and unclear topological structure (e.g., not typical tower and umbrella shapes). The model PointNet extracts the feature of each independent point and the feature of the global point cloud, and it is difficult to learn the conjunction feature from two different objects, which likely resulted in incorrect identification with incomplete canopy shapes after extracting point cloud feature from a voxel acquired by segmentation. When a voxel contains parts of multiple tree data with a bimodal distribution (i.e., a complete tree crown and a small portion (<20%) of an adjacent tree crown), the model will learn the complete information generally and always identify whole point clouds in a voxel as tree.

For the monastery garden, the spatial tree shape is a geometrical primitive with the phenotypic feature like a major trunk supporting an elliptical or conical-like shaped tree crown, which differs from the rigid objects such as buildings with regular phonotypical traits. When a voxel contains both parts of trees and buildings, the assumed errors raised when the voxel containing both tree and the wall of the temples were easily misjudged due to ambiguous phenotypic features. The point clouds in the mixture of trees and buildings are always identified as a non-tree, a reasonable explanation is that high data complexity deteriorates the useful information extracted from the tree by a deep learning network and makes the classification results of the point clouds in the voxel uncertain. The classification accuracy for the mixture point clouds in a voxel might be affected by the proportion of point clouds regarding the tree in a voxel and feature extraction means of machine learning. In contrast, a good performance was achieved for the case of the section of the buildings after voxelization. We expect that the main reason for this result is that the temples have regular surface traits different from the tree and the first T-Net in the network generated an affine transformation matrix to normalize the rotation, translation and other changes of the point cloud, which affords the efficient spatial and distance metric from multi-viewing angles and captures the globe and local features matching the semantic features corresponding to the training samples.

For the mixed forest plot with a variety of crown shapes and clustered and interwoven foliage clumps, which yields the uneven density of the forest stem distribution and overlapping shielding between crowns (Figure 7(c2)). The rich biomass forest creates complicated and difficult-to-distinguish LiDAR point patterns and deteriorates the recognition ability of the deep learning network. Hence, the point clouds regarding the overlapping trees contained in a few voxels were falsely recognized as tree. Besides, the point clouds of some tree crowns with skewed trunks and tilted tree body were not properly recognized, which differs from the upward structure of tree crown with roughly symmetric dispersed branching structures and prone to misclassification.

For the defoliated forest containing trees without leaves, the classification results are shown in Figure 7d. In the period of dormancy, trees with bare branches present deficient in the foliage elements. Judged by the globe structure of tree models, many tree skeletons were successfully recognized. However, some cases were still failed to identify by the network, e.g., a few of trees with lower parts covered by surrounding shrubs, many trees with incomplete trunk or branches cut away by adjacent voxels. Moreover, the lack of adequate training samples of defoliated trees also diminished the recognition strength of the deep learning network.

After the voxel classification based on the PointNet model, the tree crown delineation was conducted using the method mentioned in Section 2.6. The extracted individual tree crown was identically color-coded and shown in Figure 8.

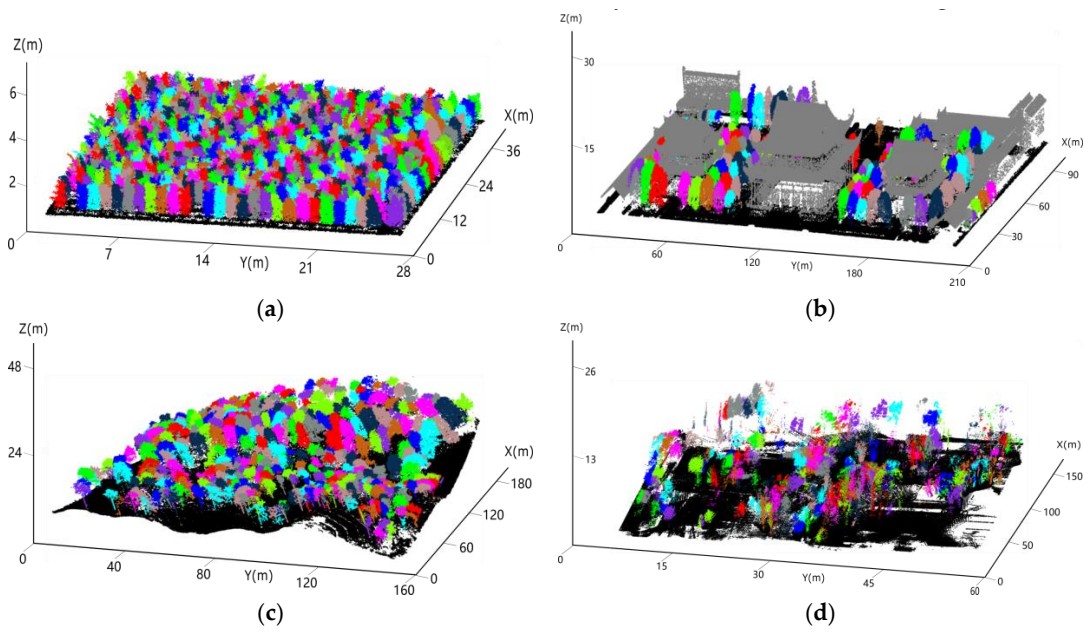

**Figure 8.** Program diagrams showing our results of individual tree crown segmentation, where different colours indicate the segmentation results for each tree. (**a–d**) show the partial segmented LiDAR data for the nursery base, monastery garden, mixed forest and defoliated forest, respectively.

Sound performance was achieved in the tree segmentation results for the four different types of forest sites using the PointNet method (Figure 8). It is found that the overall segmentation accuracy of the nursery base and the monastery garden ($r = 0.90$ and $r = 0.85$, respectively) (Table 3) are higher than that of mixed forest and defoliated forest ($r = 0.80$ and $r = 0.82$, respectively). One explanation for this difference is that the nursery base has similar tree ages, uniform planting arrangement, fewer cross-growing branches of trees and almost no understorey vegetation, which makes the voxel contain more complete tree point clouds with certain morphological characteristics. The monastery garden contains many trees with nearly spatial isolated crowns and modified shapes by manually pruning. Hence, some trees have compact envelop of crowns and are convenient for the use of height-related gradient information to realize the individual tree segmentation. Different from the nursery base, the mixed and defoliated natural forests are composed of versatile tree species and shrub compositions with interlacing and protruding branches. It is difficult for the deep learning model and gradient-based segmentation method to segment trees among the forest canopy with data deficiency caused by occlusion and other trees in the period of dormancy with bare branches and unsmoothed outmost appearance, which result in relatively poor individual tree segmentation results ($r = 0.80$ for mixed forest and $r = 0.82$ for defoliated forest, respectively). For the four types of the forest plots, some commission errors arose due to multi-foliage clumps belonging to the same tree crown, strong lateral branches generating locally convex points and the upturned eaves on the roof corner of the temples mistakenly identified as treetops.

**Table 3.** Accuracy assessments of the individual tree segmentation on the four testing sites.

|  | *NT* | *NS* | *TP* | *FP* | *FN* | *r* | *P* | *F* |
|---|---|---|---|---|---|---|---|---|
| **Nursery base** | 522 | 511 | 470 | 41 | 52 | 0.90 | 0.92 | 0.91 |
| **Monastery garden** | 160 | 151 | 136 | 15 | 24 | 0.85 | 0.90 | 0.87 |
| **Mixed forest** | 456 | 445 | 365 | 80 | 91 | 0.80 | 0.82 | 0.81 |
| **Defoliatedforest** | 167 | 163 | 137 | 26 | 30 | 0.82 | 0.84 | 0.83 |
| **Overall** | 1305 | 1270 | 1108 | 162 | 197 | 0.85 | 0.87 | 0.86 |

*NT*: the number of the trees in the plot. *NS*: the number of segmented trees. *TP*: the number of trees that were correctly segmented. *FN*: the number of segmented trees that were not detected. *FP*: the number of segmented trees that did not existing in reality but were incorrectly added by our model. *r* (recall): tree detection rate. *P* (precision): the correctness of the detected tree. *F* (F-score): the overall accuracy of the detected tree.

### 3.2. Accuracy of Tree Crown Width Estimation

For the segmented individual trees by our method, 100 trees were selected from each testing site to calculate the tree crown width in the north-south ($Cb_n$) and east-west direction ($Cb_e$) compared with manually segmented results. Correlation of coefficients ($R^2$), root mean squared error (RMSE), and relative root mean square error (rRMSE) were also calculated to evaluate the qualitative aspects of our results.

For the four testing sites, the nursery base achieves the highest accuracy of tree crown width estimation ($R^2$ = 94.4 ± 0.28%, RMSE = 0.13 ± 0.01m and rRMSE = 9.59 ± 0.70%) (Figure 9), which might be attribute to the regular and uniform geometry of tree crowns with less intersection of branches. A relatively lower accuracy was obtained for the mixed forest ($R^2$ = 85.105 ± 0.015%, RMSE = 0.74 ± 0.01m and rRMSE = 10.835 ± 0.245%) and monastery garden ($R^2$ = 88.665 ± 0.285%, RMSE = 0.57 ± 0.01m and rRMSE = 9.31 ± 0.33%), an reasonable explanation is that part of the tree canopy is blocked by surrounding tall trees or buildings, resulting in the deviation in the tree crown breadth estimation for some suppressed trees in the middle of the forest or some trees right next to the buildings. The alignment of crown width estimation between our method and manually measurements was further reduced for the defoliated forest plot. Due to many trees in the plot with bare branches and without foliage, many tree crowns have no continuous dripline and the smoothed crown surface, which leads to the generated DSM having empty cells or gaps where elevation data is missing. These detrimental factors unfavorably impacted the gradient calculation and crown width measurements. Hence, a relatively lower statistical index ($R^2$ = 79.94 ± 0.13%, RMSE = 0.61 ± 0.02m and rRMSE = 11.7 ± 0.35%) of the crown width estimation was obtained for the last plot.

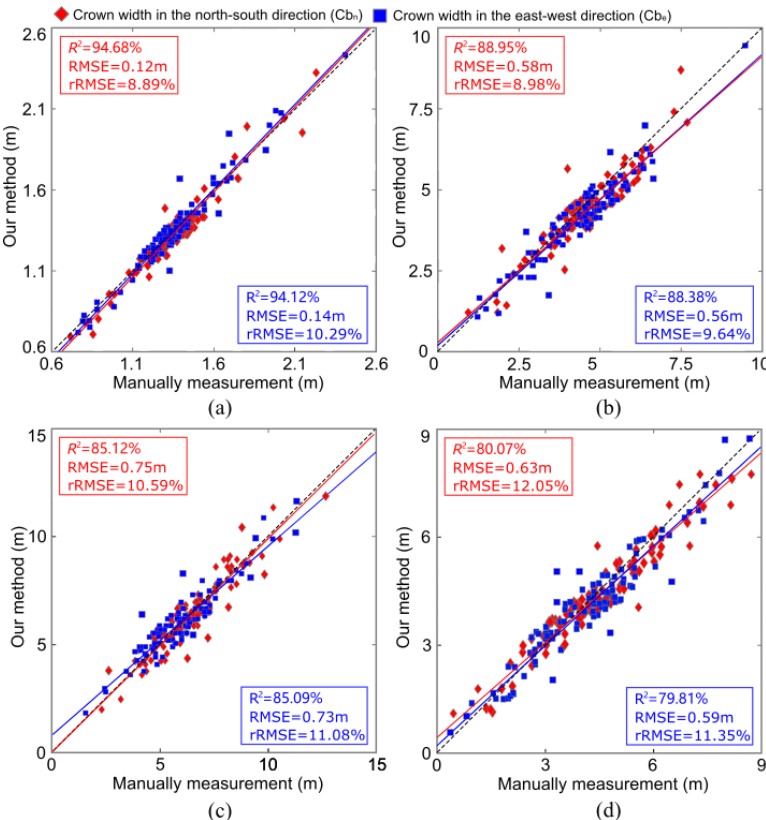

**Figure 9.** Scatter plots illustrating the comparison results of the crown width obtained by field measurements versus our method for the four different forest types, i.e., (**a**) nursery base, (**b**) monastery garden, (**c**) mixed forest, and (**d**) defoliated forests.

## 4. Discussion

### 4.1. The Advantages of Our Approach

The automatic extraction (segmentation) of individual trees from airborne laser scanning data is an important prerequisite for tree phenotypic and biophysical parameter estimation [48]. At present, machine vision algorithm and image processing techniques have been widely used in individual tree segmentation. However, it is difficult to process cluster trees with similar height and varying density distributions when only the limited geometric spatial information was incorporated. For example, the clumped tree crowns with similar heights and tight distribution may be mistakenly detected as a single treetop and leads to under-segmentation. Moreover, the non-treetop local maxima may be falsely detected as treetops and results in over-segmentation. For the segmentation of individual trees based on the centers of tree crowns and point density distributions [49], a bias is expected when trees extend one-sided tree crown or tilted tree body suffering from competitive growth with neighboring trees or environmental influences, e.g., hurricane damage or inhomogeneous distribution of solar irradiance. Clusters of the high density scanned points often appeared within the conjunction areas of the overlapping tree crowns, open branches with dense leaves or needless and occlusion-free vegetative elements exposed to the laser scanning sensors. Therefore, these issues will cause a decrease in individual tree segmentation merely depend on the limited features of point clouds.

Deep learning, which attempts to model high-level abstractions in data using a hierarchal manner, has provided machines with a greater ability to identify the target through extracting efficient features from vast samples and repeatedly improving the neural network performance [21]. In addition, with the rapid development of deep learning, a large body of research has been committed to a variety of deep learning classification or segmentation tasks using 2D images as the raw input data to realize the individual tree segmentation [50]. Although these methods have achieved good performance in tree crown segmentation, they still lose the original 3D geometric information of the studied targets in the process of being transformed into a 2D image. The disorder, non-uniformity, irregularity, and noise of forest point clouds introduce significant challenges into point cloud segmentation, and the existing image classification and segmentation framework cannot be directly applied to point cloud. Hence, we proposed a novel deep learning method of PointNet that directly processes out-of-order point cloud data to achieve the segmentation of individual trees. To the best of our knowledge, this paper is a bold attempt to employ PointNet for individual tree crown segmentation directly acting on the scanned data, which retains the spatial features of the point cloud to the greatest extent and achieves sound performance in the final test. The T-Net of the model is used to normalize the rotation, translation and other changes of scanned data in the input voxel, and The MLP of the model is used to extract numerous features from various neural networks and aggregate these features to effectively learn the characteristics of the entities regarding trees and other objects. The PointNet model in tandem with a larger number of the collected training samples obtained the optimal weights by iterative forward and back propagation in the training process, which makes the model robust to recognize the point clouds making up the tree structure.

### 4.2. Comparison with Existing Methods

The synergetic use of the voxelization strategy, the PointNet model, and the height-related gradient information on the raw point clouds were employed in our study, which is different from some existing ITC segmentation methods, such as the watershed algorithm and point cloud-based cluster segmentation algorithm.

The watershed algorithm is based on the physical principle of the asymptotic water expansion on the DSM or CHM and finally stop in the low-lying area of tree crown boundaries. However, the watershed algorithm is limited to tree species with regular shapes, which has good performance on the similar phenotypic characteristics of tree crowns, i.e., the trees neatly arranged with commonly tower or umbrella shapes. For forest

areas with irregular crown shapes and complex internals, the high canopy density of the forest and overlapping shielding between the crowns may lead to multiple local apices in the forest canopy. In addition, the performance of the watershed algorithm is prone to inappropriate dealing with the weak edge (i.e., the subtle greyscale changes on the surface of the forest appearance) and the noise on the DSM, which will produce over- and under-segmentation. These situations will be exacerbated for the watershed algorithm when encountering lush forest habitats composed of canopy and sub-canopy trees make up the multi-layered forest components with interlocked crowns and mixture species.

The point cloud-based cluster segmentation algorithm is an algorithm that adopts a top-down region growing approach to sequentially segment individually trees from the tallest to the shortest. In general, it is assumed that to analyze the geometric spatial features of the scanned points to seek apices of the tree crowns, which couples with various distance metrics to realize the individual tree segmentation. However, the key parameter of the method is uncertain for various forest plot types. If inappropriate parameter values are assigned, trees with elongated branches and serious bending branches may be over-divided, or the adjacent trees with crown overlapping may be falsely segmented. Hence, the suitable parameters are vital for the final performance of the method. Furthermore, this algorithm makes use of the 3D structure inherent in the Lidar point cloud, so mis-segmentations may occur where the canopy is unequally sampled by the laser pulses due to mutually occluded vegetative elements and varying scanning angles of the instruments.

Here, the comparison results of the watershed algorithm, point cloud-based cluster segmentation algorithm and our deep learning-based method were applied on the collected point clouds of the same four experimental forest sites (i.e., nursery base, monastery garden, mixed forest, and defoliated forest), the accuracies of the three methods are listed in Table 4. This table shows that for the nursery base with similar tree crown shape, lower planting density and neat arrangement, the three methods exhibited the similar segmentation accuracies. For the complex forests contains a broader mixture of tree species and diverse structure of trees, a small increase in the accuracy of tree segmentation was achieved, which illustrates that our deep learning framework performed better to extract spatially explicit traits of tree body when working with highly complex forest scenarios.

**Table 4.** Comparison of the accuracies of ITC segmentation using watershed algorithm, point cloud-based cluster segmentation algorithm and our method on raw point clouds of the same four experimental forest sites.

| Method | Experimental Forest Plots | NT/NS | TP | FP | FN | r | P | F |
|---|---|---|---|---|---|---|---|---|
| | Nursery base | 522/534 | 470 | 64 | 52 | 0.90 | 0.88 | 0.89 |
| | Monastery garden | 160/156 | 134 | 22 | 26 | 0.84 | 0.86 | 0.85 |
| Watershed algorithm | Mixed forest | 456/451 | 365 | 86 | 91 | 0.80 | 0.81 | 0.80 |
| | Defoliated forest | 167/165 | 127 | 38 | 40 | 0.76 | 0.77 | 0.76 |
| | Overall | 1305/1306 | 1096 | 210 | 209 | 0.84 | 0.84 | 0.84 |
| | Nursery base | 522/517 | 465 | 52 | 57 | 0.89 | 0.90 | 0.89 |
| Point cloud-based | Monastery garden | 160/147 | 134 | 13 | 26 | 0.84 | 0.91 | 0.87 |
| cluster segmentation | Mixed forest | 456/439 | 351 | 88 | 105 | 0.77 | 0.80 | 0.78 |
| algorithm | Defoliated forest | 167/169 | 127 | 42 | 40 | 0.76 | 0.75 | 0.75 |
| | Overall | 1305/1272 | 1077 | 195 | 228 | 0.83 | 0.85 | 0.84 |
| | Nursery base | 522/511 | 470 | 41 | 52 | 0.90 | 0.92 | 0.91 |
| | Monastery garden | 160/151 | 136 | 15 | 24 | 0.85 | 0.90 | 0.87 |
| Our method | Mixed forest | 456/445 | 365 | 80 | 91 | 0.80 | 0.82 | 0.81 |
| | Defoliated forest | 167/163 | 137 | 26 | 30 | 0.82 | 0.84 | 0.83 |
| | Overall | 1305/1270 | 1108 | 162 | 197 | 0.85 | 0.87 | 0.86 |

*NT*: the number of the trees. *NS*: the number of segmented trees. *r* (recall): tree detection rate. *P* (precision): the correctness of the detected tree. *F* (F-score): the overall accuracy of the detected tree. *TP*: the number of trees that were correctly segmented. *FN*: the number of segmented trees that were not detected. *FP*: the number of segmented trees that did not existing in reality but were incorrectly added by our model.

*4.3. Potential Improvements*

As mentioned in Section 3.1, it is crucial to set the appropriate voxel size in this experiment. An overly larger voxel size will make more point clouds of several objects containing in one voxel and impair machine's ability to understand semantic features of single tree. Conversely, an overly smaller voxel size will fragmentize the complete point clouds set of a single tree and adversely affect exploiting geometric knowledge of tree crowns. Here, we set the voxel size for each experiment site as the average crown width obtained from preliminary forest survey. Trees with large or small tree crowns will be the main challenges for our method. This problem is similar to the selection of the filter size for CHM smoothing prior to the water expansion using the marker controlled watershed method [51]. Some studies employed the semi-variogram statistics [51] to determine the local range of crown sizes from the CHM before individual tree crown segmentation. Likewise, strategy of automatic adoptive voxel size assignment can be further designed applied to the deep learning framework to optimize the parameter settings. In addition, as mentioned in Section 2.3, the number of point clouds in a voxel was sampled by random sampling method to 1024, which is smaller than the original number of the scanned points. In the experiment, different sampling strategy can be designed to generate more training samples from the original collected scanned points of the same tree. Meanwhile, subtle jittering of the position of each point with expected tolerance is also an alternative manner for realizing data augmentation for the training samples.

The deep learning network of PointNet only learns the local features of each point and ignores the connection relationship between points, i.e., it cannot capture the local structure induced by the metric space points live in, therefore making it unlikely to be able to learn fine-grained patterns or to understand complex scenes. Therefore, compared with the revolutionized neural networks, such as PointNet++ [52] with a class pyramid feature aggregation scheme, the ability of PointNet to explore the inter-relationship between features is slightly weaker. Further, we will combine advanced neural networks to optimize the efficiency of deep learning model and to achieve high accuracy of tree crown recognition.

## 5. Conclusions

In this paper, a deep-learning method based on the scanned point clouds collected by UAV-borne LiDAR was designed to recognize trees at voxel scale and combine the height-related gradient information to accomplish individual tree crown delineation. The proposed segmentation algorithm is composed of two stages. In the first stage, point clouds of various forms of trees and buildings were manually extracted as the training samples, which were brought into the PointNet model to train the network and obtain the optimal network parameters. Then, the point clouds of each forest sites were subdivided based on the voxelization. The point clouds in each voxel were taken as the testing samples, which was analyzed by the trained PointNet network to obtain the classification results. In the second stage, based on segmentation results of deep learning at voxel scale, a height-related gradient information was adopted to accurately depict the boundaries of each tree crown. Meanwhile, the tree crown breadth estimated from our deep learning method was compared with the manually measured results to verify the effectiveness of our approach. For the studied four forest plot types, i.e., the nursery base, the monastery garden, the mixed forest and the defoliated forest, the results revealed the best performance for the nursery base (tree crown detection rate $r = 0.90$ and crown breadth estimation $R^2 > 0.94$). For the monastery garden and mixed forest with a complex forest structure, complicated intersection of branches and different types of buildings, a sound performance was also achieved with $r = 0.85$ and $R^2 > 0.88$ for the monastery garden and $r = 0.80$, and $R^2 > 0.85$ for the mixed forest. For the fourth forest plot type with the distribution of crown defoliation across the woodland, we achieved the performance with $r = 0.82$ and $R^2 > 0.79$ for the defoliated forest. Compared with the watershed algorithm and point cloud-based cluster segmentation algorithm, the proposed method improves the tree detection accuracy by 1%–6%. Overall, this work manifested that the application of deep learning framework

directly processing on the scanned points of various forest types is feasible to solve the individual tree segmentation problem.

**Author Contributions:** Conceptualization, X.C.; Data curation, X.C., Y.Z. and T.Y.; Investigation, T.Y.; Methodology, X.C., K.J. and T.Y.; Project administration, T.Y. and X.W.; Resources, T.Y. and X.W.; Validation, X.C. and K.J.; Writing–original draft, X.C.; Writing–review & editing, X.C. and T.Y. All authors have read and agreed to the published version of the manuscript.

**Funding:** This research was funded by the National Natural Science Foundation of China (grant number 31770591 and 41701510). This research was also financially supported by the Central Public-interest Scientific Institution Basal Research Fund for Chinese Academy of Tropical Agricultural Sciences (grant number 1630022020002).

**Conflicts of Interest:** The authors declare no conflict of interest.

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
