# Peer review of "Individual Tree Crown Segmentation Directly from UAV-Borne LiDAR Data Using the PointNet of Deep Learning"

_forests, doi:10.3390/f12020131_

Round 1
Reviewer 1 Report
The applied number of training samples seems to be a little limited, even with augmentation, so let me raise some questions/suggestions related to augmentation:
Have you applied non-z-axis based rotation (tilting) as well? What was the maximum angle for tilting? This might be a tree type dependent parameter and might affect gound removal. I suggest explaining augmentation in more detail.
If I understand correctly, voxelization was made totally separated and not included in the training. So augmentation is done only within a voxel. Do you see any chance to include voxelization in augmentation? I would suggest mentioning this in the paper whether it's already included in augmentation or as further work.
I agree that voxel size is a crucial point; this worth further investigation. I see that there should be an automated way to determine voxel size based on tree type, vegetation density, type of the region (urban area, forest, etc.).
If I understand correctly, changing the voxel size implies retraining the network (~100 hours). If voxel size determination is a manual fine-tuning step, this kind of segmentation will be hard to use in practice.
I suggest explaining your plans for improving voxel size determination.
Author Response
1、Have you applied non-z-axis based rotation (tilting) as well? What was the maximum angle for tilting? This might be a tree type dependent parameter and might affect gound removal. I suggest explaining augmentation in more detail.
Answer: Thanks for your valuable suggestion. We remove the original unclear phrasing and rewrote the method of data augmentation, which was shown before Figure 3.
The specific rewritten paragraph is as follows:“We generated new training data set based on the rotation of the entire point cloud in each voxel by a random angle and along the vertical axis. Meanwhile, the strategy of moving every point in each voxel with a small offset along a random vector, i.e., jittering the position of each point of every training sample by a Gaussian noise with zero mean and a small standard deviation (ranging 0.02-0.06)”.
2、If I understand correctly, voxelization was made totally separated and not included in the training. So augmentation is done only within a voxel. Do you see any chance to include voxelization in augmentation? I would suggest mentioning this in the paper whether it's already included in augmentation or as further work.
Answer: Thank you for the comments. The point clouds of forest plot was discretized into 3D voxels with a given resolution. The aim of defining a reasonable voxel resolution is to give an appropriate presentation of each tree in every voxel. We still not find the correlations between the voxelization and data augmentation elaborated in the existing literatures. In the first paragraph of sub section 4.3, we illustrated the related situations regarding this comments, which is also listed as follows:
In addition, as mentioned in 2.3, the number of point clouds in a voxel was sampled by random sampling method to 1024, which is smaller than the original number of the scanned points. In the experiment, different sampling strategy can be designed to generate more training samples from the original collected scanned points of the same tree. Meanwhile, subtle jittering of the position of each point with expected tolerance is also an alternative manner for realizing data augmentation for the training samples.
3、I agree that voxel size is a crucial point; this worth further investigation. I see that there should be an automated way to determine voxel size based on tree type, vegetation density, type of the region (urban area, forest, etc.). If I understand correctly, changing the voxel size implies retraining the network (~100 hours). If voxel size determination is a manual fine-tuning step, this kind of segmentation will be hard to use in practice. I suggest explaining your plans for improving voxel size determination.
Answer: Thank you for the valuable comments. As mentioned in section 3.1, it is crucial to set the appropriate voxel size in this experiment. An overly larger voxel size will make more point clouds of several objects containing in one voxel, and impair machine’s ability to understand semantic features of single tree. Conversely, an overly smaller voxel size will fragmentize the complete point clouds set of a single tree and adversely affect exploiting geometric knowledge of tree crowns. Here, we set the voxel size for each experiment site as the average crown width obtained from preliminary forest survey. Trees with large or small tree crowns will be the main challenges for our method. This problem is similar to the selection of the filter size for CHM smoothing prior to the water expansion using the marker controlled watershed method[51]. Some studies employed the semi-variogram statistics[51] to determine the local range of crown sizes from the CHM before individual tree crown segmentation. Likewise, strategy of automatic adoptive voxel size assignment can be further designed applied to the deep learning framework to optimize the parameter settings.
The above-mentioned contents were added into subsection 4.3.
- Hu, B.; Li, J.; Jing, L.; Judah, A. Improving the efficiency and accuracy of individual tree crowndelineation from high-density LiDAR data. Int. J. Appl. Earth Obs. Geoinf.2014, 26, 145–155, doi:10.1016/j.jag.2013.06.003.

Reviewer 2 Report
The Introduction section is too long in the level of existing mehods description. All the informations are important but please reduce the size of the text.
My main substantive doubt/suggestion:
Why was the performance made only for the leafy period? In many regions of the World lidar data are available, but often only for the winter period. It would be better to test segmentation methods to show the differences in ITC segmetation resulting from the different properties of vegetation - with and without leaves. If it is not possible to change the approach in your work considering my suggestion, at least, please in the discussion section refer to the literature presenting this point of view which is widely mentioned.
Author Response
1、The Introduction section is too long in the level of existing methods description. All the informations are important but please reduce the size of the text.
Answer: We have deleted some repetitive phrases describing the deep learning models in the Introduction section (the fourth paragraph) according to your suggestions. Please review these changes marked in blue in the revised version.
2、Why was the performance made only for the leafy period? In many regions of the World lidar data are available, but often only for the winter period. It would be better to test segmentation methods to show the differences in ITC segmetation resulting from the different properties of vegetation - with and without leaves. If it is not possible to change the approach in your work considering my suggestion, at least, please in the discussion section refer to the literature presenting this point of view which is widely mentioned.
Answer: According to your comments, we have added one experiment of the defoliated forest plot processing using our deep learning framework. The training and testing samples of the defoliated trees were collected and many figures were redrawn accordingly. The quantitative results were added into Table 3 and 4, and corresponding analysis were elaborated in each section. Please review these changes marked in blue shown in the revised version.

Reviewer 3 Report
This is a good and very thorough paper. I liked how thoroughly and clearly you explain your method. I don’t have any major issues, however, I do have a long list of minor corrections that I think need to be addressed: confusing sentence construction, sections in the wrong place.... See list below.
Ln15 “directly processing”?
Ln 16-20 Sentence confusingly structured. Maybe “The specific steps of our approach were as follows: first, a voxelisation strategy was conducted to subdivide collected point clouds regarding various tree species of various forest types into many voxels. These voxels were taken as training samples of the PointNet deep learning framework to recognise the tree crowns at the voxel scale.”
Ln17 “voxelisation strategy of our approach” delete “of our approach”
Ln 28 “types of buildings existing” delete existing
Ln 28 “a sound performance”
Ln41 especially in tropical regions,
Ln44 “freedom from the constraints of light and climate conditions” climate conditions is not really accurate for all LIDAR – example fog or thick cloud?
Ln44-45, has led to it becoming one of the most efficient….
Ln 50 “which can largely lower” delete “largely”
Ln60 “always impacted” caused?
Ln65 “ecological forests” do you mean natural forests?
Ln82 classification tasks
Ln85 “The method based on voxelisation can be effectively maintained the spatial relationship between voxels.” Don’t understand – reword/clarify
LN102 delete “delicately”
Fig1 Abbreviation DSM not yet given
Fig1 Not familiar with “grident”. Is this a mis-spelling?
Ln128 Delete “In addition”
Ln164-165 Sentence “The LiDAR point density values were……respectively.” Respective to what? Presumably to the three forest plots but you don’t say
Ln165-166 Do you mean “The number of trees in the training subset of experimental sites 1,2 and 3 were ….”
Ln 177 “mixed forest ^were^ used to test…”
Ln226 “Two “shared” in figure 4” Not entirely sure what you are referring to here - clarify
Ln 277 Delete “continuing”
Ln278 “an underrepresented class ^which^ are much more difficult”
Ln300 PointNet misspelled
Ln308-309 “and about 25% of the sub-canopy trees” Do you mean 25% of the trees were sub-canopy?
Ln315-351 Very long paragraph – break it up into sections, for the different test sites
Ln317 “when the voxel containing scanned points was regarded as a small portion of tree crowns with topological structure vagueness” Don’t understand this point – reword and clarify
Ln 317 “or the voxel does not have to contain vertices of treetops.” Don’t understand – you mean voxel doesn’t contain treetop vertices?
Ln325 “In contrast”
Ln327 “”The proportion of a tree crown in a voxel larger than 75% always recognized as a tree” Don’t understand – you mean if the tree crown proportion contained in a voxel is greater than 75% it is always recognized as a tree?
Ln329 “voxel containing both tree and the wall”
Ln 331 3D Tree model – what is this?
Ln 335 “identify it as a non-tree”
Ln335 “is likely that ^for^ the point cloud…”
Ln337 “judged as the non-tree” delete “the”
Ln349 “and leads the model PointNet” change: maybe “and so the model PointNet…”
Ln 351 “excluded from the voxel, which caused misclassification.”
Ln 353 Section 3.2 Surely this section belongs in Section 2, the methods section. You shouldn’t be introducing equations and explaining your method in the Results.
Ln 378 “tree crown information ^and^ lead to large…” ?
Ln 387 “voxel, which in order” delete which
Ln406”number of segmented trees that don’t exist in reality but are incorrectly added by our model.”
Ln432 “In addition, ^for the^ nursery base…”
Ln434 “almost no small trees or understory vegetation”
Ln435 accurate would be better than “well”
Ln 437 “recognize this type of voxel”
Ln438 “some trees have overlapping shielding between the crowns, and the use of height-related gradient information to realise the individual tree segmentation causes commission errors.”
Ln441 what are “versatile trees”?
Ln 441 “shrub compositions can be falsely identified as part…”?
Ln443 “causing this part of the point cloud to be discarded”
Ln452-453you mean “FP: the number of segmented trees that don’t exist in reality and are incorrectly identified”
Ln466 “a sound performance”
Ln 468 “which might be attributed to the different types…” What might be attributed?
Ln469-475 I think you might be better to give the results for Test Site 3, then explain why the accuracy is poorer
Ln470-472 “…may vary in size and shape, ^the^ structure and composition of tree canopies can be highly complex due to the intersection…”
Ln473 “biased results” less accurate results than for the other test sites?
Ln475 Delete “It is likely not difficult to find that”
Ln493”prerequisite for estimation tree” delete estimation
Ln 499 Delete “inevitably”
Ln 503 “These issues would cause a marked decrease in ITC accuracy”
Ln506 “has provided machines ^with^ a greater”
Ln508 “large body of research”
Ln516 “to the best of our knowledge, this paper is the first time PointNet has been employed for individual tree crown segmentation.”
Ln522 “model achieve better”
Ln522 “MLP has ^been^ used to aggregate…”?
Ln525-558 this is an lengthy, intimidating paragraph. Maybe break up into seperate paragraphs
Ln532-535 “For forest areas with irregular crown shapes and complex internals, the high canopy density of the forest and overlapping shielding between the crowns, may lead to multiple local extremums in the same canopy.”
Ln535 Delete “easily”
Ln 535 “has a good performance to the weak edge” not sure what this “weak edge” means
Ln 538 “loss of precision of the crown boundary will be significantly improved” Do you mean “will significantly worsen” –
Ln545 “serious bending branches” what are these – either explain or change
Ln556 “severe bending of the tree” again, not entirely sure what this means – explain or change
Ln564-565 you mean “FP: the number of segmented trees that don’t exist in reality and are incorrectly identified”
Ln573 “follow-up work of this experiment” to “follow-up work to this experiment”
Ln599 Delete “existing”
Ln599 change “the sound performance” with “a sound performance”
Author Response
1、Ln15 “directly processing”?
Answer:Accepted. We modified the “directly processes” to “directly processing(line 17 of the revised version)” .
2、Ln 16-20 Sentence confusingly structured. Maybe “The specific steps of our approach were as follows: first, a voxelisation strategy was conducted to subdivide collected point clouds regarding various tree species of various forest types into many voxels. These voxels were taken as training samples of the PointNet deep learning framework to recognise the tree crowns at the voxel scale.”
Answer: Thanks a lot. We rewrote the “The specific steps of our approach were as follows...(original line 16-20)” according to your suggestions. Please see the corresponding text in the revised version.
3、Ln17 “voxelisation strategy of our approach” delete “of our approach”
Answer: Accepted. We deleted the “of our approach(line 20 of the revised version)” .
4、Ln 28 “types of buildings existing” delete existing
Answer: Accepted. We deleted the “existing(line 30 of the revised version)” .
5、Ln 28 “a sound performance”
Answer: Accepted. We modified the “the sound performance” to “a sound performance(line 28 of the revised version)” .
6、Ln41 especially in tropical regions,
Answer: Accepted. We rewrote the sentence “Forest parameters, such as tree location...(original line 41)” . Please see the corresponding text in the revised version.
7、Ln44 “freedom from the constraints of light and climate conditions” climate conditions is not really accurate for all LIDAR – example fog or thick cloud?
Answer: Accepted. We modified the “Light detection and ranging (LiDAR) is an active remote sensing technology...(original lines 43-45)” to “Light detection and ranging (LiDAR) is an active remote sensing technology, as its high precision and high efficiency, has led to it becoming one of the most efficient surveying techniques for acquiring detailed and accurate target phenotypic data [4](lines 46-48 of the revised version)” .
8、Ln44-45, has led to it becoming one of the most efficient….
Answer: Accepted. We rewrote the sentence “Light detection and ranging (LiDAR) is an active remote sensing technology...(original lines 44-45)” . Please see the corresponding text in the revised version(lines 46-48 of the revised version).
9、Ln 50 “which can largely lower” delete “largely”
Answer: Accepted. We deleted the “largely(line 52 of the revised version)” .
10、Ln60 “always impacted” caused?
Answer: Accepted. We modified the “impacted” to “caused(line 63 of the revised version)” .
11、Ln65 “ecological forests” do you mean natural forests?
Answer: Accepted. We modified the “ecological forests” to “natural forests(line 68 of the revised version)” .
12、Ln82 classification tasks
Answer: Accepted. We modified the “classification task” to “classification tasks(line 86 of the revised version)” .
13、Ln85 “The method based on voxelization can be effectively maintained the spatial relationship between voxels.” Don’t understand – reword/clarify
Answer: Accepted. We rewrote the sentence follows: “The method based on voxelization can be effectively retained the original spatial information of the point clouds in each voxels, which is beneficial for subsequent refinement processing for the accurate target depiction”. Please review these changes in the revised version (the fourth paragraph of the introduction).
14、LN102 delete “delicately”
Answer: Accepted. We deleted the “delicately(line 105 of the revised version)”.
15、Fig1 Abbreviation DSM not yet given
Answer: Accepted. We have given the full name of DSM and modified Figure 1.
16、Fig1 Not familiar with “grident”. Is this a mis-spelling?
Answer: Accepted. We corrected the “grident” to “gradient” and modified Figure 1. .
17、Ln128 Delete “In addition”
Answer: Accepted. We deleted the “In addition(line 131 of the revised version)”.
18、Ln164-165 Sentence “The LiDAR point density values were……respectively.” Respective to what? Presumably to the three forest plots but you don’t say
Answer: Accepted. We rewrote the section. Please see the corresponding text in the revised version(lines 166-175 of the revised version).
19、Ln165-166 Do you mean “The number of trees in the training subset of experimental sites 1,2 and 3 were ….”
Answer: Accepted. We rewrote the sentences as follows: “The number of training samples (trees and buildings) for the nursery base, monastery garden, mixed forest plot and defoliated forest landscapes were 501 (trees), 168(trees) / 334(buildings), 426(trees) and 166(trees), respectively”.
20、Ln 177 “mixed forest ^were^ used to test…”
Answer: Accepted. We modified the “mixed forest used to test” to “mixed forest were used to test(line 187 of the revised version)” .
21、Ln226 “Two “shared” in figure 4” Not entirely sure what you are referring to here - clarify
Answer: Accepted. We rewrote the sentence as follows: “The aligned data of point clouds () in each voxels was passed through a multi-layer perceptron (MLP (64, 64)) with the given numbers of layer sizes shown in the bracket to obtain the matrix (). The fully connected layers of MLP are shown by the three dotted boxes in the upper part of Figure 3”, which is show in the last paragraph of subsection 2.4.
22、Ln 277 Delete “continuing”
Answer: Accepted. We deleted the “continuing(line 342 of the revised version)”.
23、Ln278 “an underrepresented class ^which^ are much more difficult”
Answer: Accepted. We modified the “During the training process continuing...(original line 278)” to “During the training process, the neuron network encountered some complicated samples in a batch, e.g., a voxel containing parts of multiple trees, small proportion of an individual tree data or some dwarf shrubs, which impaired the learning efficacy of the model and resulted in the strong fluctuations in the value of the regression loss function(lines 342-345 of the revised version)” .
24、Ln300 PointNet misspelled
Answer: Accepted. We corrected the “PonitNet” to “PointNet(line 367 of the revised version) ” .
25、Ln308-309 “and about 20% of the sub-canopy trees” Do you mean 20% of the trees were sub-canopy?
Answer: Accepted. We modified the “For the mixed forest (testing site 3)...(original lines 308-309) ” to “For the mixed forest (testing site 3) with various sizes of tree crowns, the complicated intersection of branches and containing roughly 15% of the sub-canopy trees, the defoliated forest (testing site 4) with bare branches and a few of trees with lower parts covered by surrounding shrubs, it is difficult to ensure that a voxel contains a complete tree(lines 374-378 of the revised version)” .
26、Ln315-351 Very long paragraph – break it up into sections, for the different test sites
Answer: Accepted. We divide this paragraph into 5 subsections. Please review these changes in the revised version(lines 386-431 of the revised version).
27、Ln317 “when the voxel containing scanned points was regarded as a small portion of tree crowns with topological structure vagueness” Don’t understand this point – reword and clarify
Answer: Accepted. We modified the “For the nursery base, as shown in Figure 6(a2)...(original line 317)” to “For the nursery base, as shown in Figure 7(a2), the main errors appeared when the voxel containing scanned points was regarded as a small portion of tree saplings with immature tree crown and unclear topological structure (e.g., not typical tower and umbrella shapes)(lines 387-390 of the revised version)” .
28、Ln 317 “or the voxel does not have to contain vertices of treetops.” Don’t understand – you mean voxel doesn’t contain treetop vertices?
Answer: Accepted. We modified the “For the nursery base, as shown in Figure 6(a2)...(original line 317)” to “For the nursery base, as shown in Figure 7(a2), the main errors appeared when the voxel containing scanned points was regarded as a small portion of tree saplings with immature tree crown and unclear topological structure (e.g., not typical tower and umbrella shapes)(lines 387-390 of the revised version)”.
29、Ln325 “In contrast”
Answer: Accepted. We delete the “In the contrast(lines 386-396 of the revised version)”.
30、Ln327 “The proportion of a tree crown in a voxel larger than 75% always recognized as a tree” Don’t understand – you mean if the tree crown proportion contained in a voxel is greater than 75% it is always recognized as a tree”?
Answer: Accepted. We modified the “The proportion of a tree crown contained in a voxel larger than 75% always recognized as a tree (Figure 6(a3, a4))” to “When a voxel contains parts of multiple tree data with a bimodal distribution (i.e., a complete tree crown and a small portion (<20%) of an adjacent tree crown), the model will learn the complete information generally and always identify whole point clouds in a voxel as tree(lines 393-396 of the revised version)” .
31、Ln329 “voxel containing both tree and the wall”
Answer: Accepted. We modified the “For the monastery garden, the main errors appeared...(original lines 329-331)” to “When a voxel contains both parts of trees and buildings, the assumed errors raised when the voxel containing both tree and the wall of the temples was easily misjudged due to ambiguous phenotypic features(lines 399-401 of the revised version)” .
32、Ln 331 3D Tree model – what is this?
Answer: Accepted. We modified the “3D Tree model is a geometrical primitive with the phenotypic feature like a major trunk conjunct with an umbrella or conical-like shape, which differs from the roof of temples” to “the spatial tree shape is a geometrical primitive with the phenotypic feature like a major trunk supporting an elliptical or conical-like shaped tree crown, which differs from the rigid objects such as buildings with regular phonotypical traits(lines 397-399 of the revised version)” .
33、Ln 335 “identify it as a non-tree”
Answer: Accepted. We modified the “identify it as the non-tree” to “identify it as a non-tree(line 402 of the revised version)” .
34、Ln335 “is likely that ^for^ the point cloud…”
Answer: Accepted. We modified the “The reason is likely that the point cloud in a voxel with high data complexity...(original lines 335-338)” to “The point clouds in mixture of trees and buildings are always identified it as a non-tree, a reasonable explanation is that high data complexity deteriorates the useful information extracted from the tree by a deep learning network and leads to the classification results of the point clouds in the voxel uncertain(lines 401-405 of the revised version)” .
35、Ln337 “judged as the non-tree” delete “the”
Answer: Accepted. We delete the “the” and modified the “The reason is likely that the point cloud...(original lines 335-338)” to “The classification accuracy for the mixture point clouds in a voxel might be affected by the proportion of point clouds regarding the tree in a voxel and feature extraction means of machine learning(lines 405-407 of the revised version)”.
36、Ln349 “and leads the model PointNet” change: maybe “and so the model PointNet…”
Answer: Accepted. We modified the “The case of the overlapping shielding between the crowns...(original lines 347-350)” to “ The rich biomass forest creates complicated and difficult-to-distinguish LiDAR point patterns and deteriorates the recognition ability of the deep learning network. Hence, the point clouds regarding the overlapping trees contained in a few voxels were falsely recognized as tree(lines 415-418 of the revised version)” .
37、Ln 351 “excluded from the voxel, which caused misclassification.”
Answer: Accepted. We modified the “Besides, the trunk or crown of some skewed trees will be excluded from the voxel, which differs from the complete and useful tree features and caused misclassification” to “Besides, the point clouds of some tree crowns with skewed trunks and tilted tree body was not properly recognized, which differs from the upward structure of tree crown with roughly symmetric dispersed branching structures and prone to misclassification(lines 418-421 of the revised version)” .
38、Ln 353 Section 3.2 Surely this section belongs in Section 2, the methods section. You shouldn’t be introducing equations and explaining your method in the Results.
Answer: Accepted. We add a new section “2.6. Individual tree segmentation” and moved the paragraphs on the results that were originally described in section 3.2 to 2.6 according to your comment. Please review these changes in the revised version.
39、Ln 378 “tree crown information ^and^ lead to large…” ?
Answer: Accepted. We add a new section “2.6. Individual tree segmentation” and moved the paragraphs on the results that were originally described in section 3.2 to 2.6 according to your comment. Then, we made some appropriate changes to this section. Please review these changes in the revised version.
40、Ln 387 “voxel, which in order” delete which
Answer: Accepted. We add a new section “2.6. Individual tree segmentation” and moved the paragraphs on the results that were originally described in section 3.2 to 2.6 according to your comment. Then, we made some appropriate changes to this section. Please review these changes in the revised version.
41、Ln406”number of segmented trees that don’t exist in reality but are incorrectly added by our model.”
Answer: Accepted. We modified the “FP was (false positive) the number of segmented trees that not existing in the region but were incorrectly added (commission error)” to “FP represents (false positive) the number of segmented trees that do not exist in reality but were incorrectly added (commission error) by our model (lines 313-315 of the revised version)” .
42、Ln432 “In addition, ^for the^ nursery base…”
Answer: Accepted. We modified the “ In addition, nursery base with similar tree age and plant spacing...(original lines 432-434)” to “One explanation for this difference is that the nursery base has similar tree ages, uniform planting arrangement, fewer cross-growing branches of trees and almost no understorey vegetation, which makes the voxel contains more complete tree point clouds with certain morphological characteristics(lines 443-446 of the revised version)” .
43、Ln434 “almost no small trees or understory vegetation”
Answer: Accepted. We modified the “In addition, nursery base with similar tree age and plant spacing...(original lines 432-434)” to “One explanation for this difference is that the nursery base has similar tree ages, uniform planting arrangement, fewer cross-growing branches of trees and almost no understorey vegetation, which makes the voxel contains more complete tree point clouds with certain morphological characteristics(lines 443-446 of the revised version)” .
44、Ln435 accurate would be better than “well”
Answer: Accepted. We delete the “Therefore, the result of individual tree segmentation is very well” .
45、Ln 437 “recognize this type of voxel”
Answer: Accepted. We modified the “The monastery garden with unique phenotypic features...(original lines 435-438)” to “ The monastery garden contains many trees with nearly spatial isolated crowns and modified shapes by manually pruning(lines 446-448 of the revised version)” .
46、Ln438 “some trees have overlapping shielding between the crowns, and the use of height-related gradient information to realise the individual tree segmentation causes commission errors.”
Answer: Accepted. We modified the “Moreover, some trees with overlapping shielding between the crowns, which cause the commission errors by using height-related gradient information to realize the individual tree segmentation” to “Hence, some trees have compact envelop of crowns and are convenient for the use of height-related gradient information to realize the individual tree segmentation(lines 448-449 of the revised version)” .
47、Ln441 what are “versatile trees”?
Answer: Accepted. We rewrote the sentence “Different from the nursery base, the mixed forest with interlacing and protruding branches, versatile trees and shrub compositions was falsely taken as part of the adjacent tree crowns” to “Different from the nursery base, the mixed and defoliated natural forests are composed of versatile tree species and shrub compositions with interlacing and protruding branches(lines 449-451 of the revised version)”.
48、Ln 441 “shrub compositions can be falsely identified as part…”?
Answer: Accepted. We modified the “Different from the nursery base, the mixed forest with interlacing and protruding branches, versatile trees and shrub compositions was falsely taken as part of the adjacent tree crowns” to “Different from the nursery base, the mixed and defoliated natural forests are composed of versatile tree species and shrub compositions with interlacing and protruding branches(lines 449-451 of the revised version)” .
49、Ln443 “causing this part of the point cloud to be discarded”
Answer: Accepted. We modified the “Moreover, it is difficult for the model to recognize the tree...(original lines 442-444)” to “It is difficult for the deep learning model and gradient-based segmentation method to segment trees among the forest canopy with data deficiency caused by occlusion and other trees in the period of dormancy with bare branches and unsmoothed outmost appearance, which result in relatively poor individual tree segmentation results (r=0.80 for mixed forest and r=0.82 for defoliated forest, respectively)(lines 451-455 of the revised version)” .
50、Ln452-453you mean “FP: the number of segmented trees that don’t exist in reality and are incorrectly identified”
Answer: Accepted. We modified the “FP: the number of segmented trees that don’t exist in reality and are incorrectly identified” to “FP: the number of segmented trees that did not existing in reality but were incorrectly added by our model(lines 462-463 of the revised version)” .
51、Ln466 “a sound performance”
Answer: Accepted. We rewrote the section 3.2 (the original section 3.3), Please review these changes in the revised version.
52、Ln 468 “which might be attributed to the different types…” What might be attributed?
Answer: Accepted. We rewrote the section 3.2 (the original section 3.3), Please review these changes in the revised version.
53、Ln469-475 I think you might be better to give the results for Test Site 3, then explain why the accuracy is poorer
Answer: Accepted. We rewrote the section 3.2 (the original section 3.3). Please review these changes in the revised version.
54、Ln470-472 “…may vary in size and shape, ^the^ structure and composition of tree canopies can be highly complex due to the intersection…”
Answer: Accepted. We rewrote the section 3.2 (the original section 3.3). Please review these changes in the revised version.
55、Ln473 “biased results” less accurate results than for the other test sites?
Answer: Accepted. We rewrote the section 3.2 (the original section 3.3). Please review these changes in the revised version.
56、Ln475 Delete “It is likely not difficult to find that”
Answer: Accepted. We rewrote the section 3.2 (the original section 3.3). Please review these changes in the revised version.
57、Ln493”prerequisite for estimation tree” delete estimation
Answer: Accepted. We deleted the “estimation(line 493 of the revised version)”.
58、Ln 499 Delete “inevitably”
Answer: Accepted. We deleted the “inevitably(line 499 of the revised version)”.
59、Ln 503 “These issues would cause a marked decrease in ITC accuracy”
Answer: Accepted. We modified the “The accuracy of ITC would markedly decrease which is caused by these issues” to “Therefore, these issues will cause a decrease in individual tree segmentation merely depend on the limited features of point clouds(lines 506-507 of the revised version)” .
60、Ln506 “has provided machines ^with^ a greater”
Answer: Accepted. We added the “with(line 509 of the revised version)”.
61、Ln508 “large body of research”
Answer: Accepted. We modified the “a large body of researches” to “a large body of research(line 511 of the revised version)” .
62、Ln516 “to the best of our knowledge, this paper is the first time PointNet has been employed for individual tree crown segmentation.”
Answer: Accepted. We modified the “To best of our knowledge, PointNet is the first employed for individual tree crown segmentation in our paper” to “ To the best of our knowledge, this paper is a bold attempt that PointNet has been employed for individual tree crown segmentation directly acting on the scanned data, which retains the spatial features of the point cloud to the greatest extent and achieves sound performance in the final test(lines 520-523 of the revised version)” .
63、Ln522 “model achieve better”
Answer: Accepted. We modified the “The T-Net of the model is used to normalize the rotation...(original lines 522-523)” to “The T-Net of the model is used to normalize the rotation, translation and other changes of scanned data in the input voxel, and The MLP of the model is used to extract numerous features from various neural networks and aggregate these features to effectively learn the characteristics of the entities regarding trees and other objects(lines 523-526 of the revised version)” .
64、Ln522 “MLP has ^been^ used to aggregate…”?
Answer: Accepted. We modified the “MLP has usually used to aggregates the high-dimensional local features of each voxel to learn the characteristics as completely as possible” to “The T-Net of the model is used to normalize the rotation, translation and other changes of scanned data in the input voxel, and The MLP of the model is used to extract numerous features from various neural networks and aggregate these features to effectively learn the characteristics of the entities regarding trees and other objects(lines 523-526 of the revised version)” .
65、Ln525-558 this is an lengthy, intimidating paragraph. Maybe break up into seperate paragraphs
Answer: Accepted. We divide this paragraph into 4 subsections. Please review these changes in the revised version(lines 531-567 of the revised version).
66、Ln532-535 “For forest areas with irregular crown shapes and complex internals, the high canopy density of the forest and overlapping shielding between the crowns, may lead to multiple local extremums in the same canopy.”
Answer: Accepted. We modified the “For forest areas with irregular crown shapes and complex internals...(original lines 532-535)” to “For forest areas with irregular crown shapes and complex internals, the high canopy density of the forest and overlapping shielding between the crowns may lead to multiple local apices in the forest canopy(lines 539-541 of the revised version)” .
67、Ln535 Delete “easily”
Answer: Accepted. We deleted the “easily(lines 541 of the revised version)”.
68、Ln 535 “has a good performance to the weak edge” not sure what this “weak edge” means
Answer: Accepted. We modified the “ In addition, the watershed algorithm has a good performance...((original lines 535-537))” to “ In addition, the performance of the watershed algorithm is prone to inappropriate dealing with the weak edge (i.e., the subtle greyscale changes on the surface of the forest appearance) and the noise on the DSM, which will produce over- and under- segmentation(lines 541-544 of the revised version)” .
69、Ln 538 “loss of precision of the crown boundary will be significantly improved” Do you mean “will significantly worsen” –
Answer: Accepted. We modified the “ Especially when the watershed algorithm is used in forests...(original lines 541-543)” to “ These situations will be exacerbated for the watershed algorithm when encountering lush forest habitats composed of canopy and sub-canopy trees make up the multi-layered forest components with interlocked crowns and mixture species(lines 544-546 of the revised version)” .
70、Ln545 “serious bending branches” what are these – either explain or change
Answer: Accepted. We rewrote the section 4.2 (the third paragraph). Please review these changes in the revised version(lines 557-557 of the revised version).
71、Ln556 “severe bending of the tree” again, not entirely sure what this means – explain or change
Answer: Accepted. We rewrote the section 4.2 (the fourth paragraph). Please review these changes in the revised version(lines 558-567 of the revised version).
72、Ln564-565 you mean “FP: the number of segmented trees that don’t exist in reality and are incorrectly identified”
Answer: Accepted. We modified the “FP: the number of segmented trees that don’t exist in reality and are incorrectly identified” to “FP: the number of segmented trees that did not existing in reality but were incorrectly added by our model(lines 574-575 of the revised version)” .
73、Ln573 “follow-up work of this experiment” to “follow-up work to this experiment”
Answer: Accepted. We delete the sentence”To address this issue, our follow-up work of this experiment will be combined with the samples of more complex experimental areas to improve the accuracy of point cloud detection”.
74、Ln599 Delete “existing”
Answer: Accepted. We deleted the “existing(line 619 of the revised version)”.
75、Ln599 change “the sound performance” with “a sound performance”
Answer: Accepted. We modified the “the sound performance” to “a sound performance(line 619 of the revised version)” .

Round 2
Reviewer 2 Report
I accept the current form of the manuscript. As a further consideration, I suggest to develop the classification of non leavy trees and shrubs using training samples collected in the field in many different cases (different species, habitats and forms).